# Recruitment of mRNAs to P granules by condensation with intrinsically-disordered proteins

Chih-Yung S Lee[1], Andrea Putnam[1], Tu Lu[1], ShuaiXin He[1,2], John Paul T Ouyang[1], Geraldine Seydoux[1]*

[1]HHMI and Department of Molecular Biology and Genetics, Johns Hopkins University School of Medicine, Baltimore, United States; [2]Department of Biophysics and Biophysical Chemistry, Johns Hopkins University School of Medicine, Baltimore, United States

**Abstract** RNA granules are protein/RNA condensates. How specific mRNAs are recruited to cytoplasmic RNA granules is not known. Here, we characterize the transcriptome and assembly of P granules, RNA granules in the *C. elegans* germ plasm. We find that P granules recruit mRNAs by condensation with the disordered protein MEG-3. MEG-3 traps mRNAs into non-dynamic condensates in vitro and binds to ~500 mRNAs in vivo in a sequence-independent manner that favors embryonic mRNAs with low ribosome coverage. Translational stress causes additional mRNAs to localize to P granules and translational activation correlates with P granule exit for two mRNAs coding for germ cell fate regulators. Localization to P granules is not required for translational repression but is required to enrich mRNAs in the germ lineage for robust germline development. Our observations reveal similarities between P granules and stress granules and identify intrinsically-disordered proteins as drivers of RNA condensation during P granule assembly.

*For correspondence:
gseydoux@jhmi.edu

## Introduction

RNA granules are RNA/protein condensates that assemble in the absence of limiting membranes. RNA granules form by phase separation, a de-mixing process that drives condensation of RNA and proteins into liquid or gel-like phases (*Kato et al., 2012*; *Weber and Brangwynne, 2012*; *Hyman et al., 2014*; *Lin et al., 2015*; *Shin and Brangwynne, 2017*; *Boeynaems et al., 2018*; *Mittag and Parker, 2018*; *Alberti et al., 2019*). Intrinsically-disordered domains in RNA-binding proteins mediate labile, multivalent protein-protein interactions that drive phase separation in vitro (*Kato et al., 2012*; *Banani et al., 2017*; *Shin and Brangwynne, 2017*; *Boeynaems et al., 2018*; *Mittag and Parker, 2018*). RNA also drives phase separation by acting as a scaffold for multivalent RNA-binding proteins or by participating in intermolecular RNA:RNA interactions (*Van Treeck and Parker, 2018*). RNA:RNA interactions can be sequence-specific (*Langdon et al., 2018*) or non-sequence specific (*Van Treeck and Parker, 2018*). Total RNA extracted from yeast cells phase separates in vitro in the absence of any proteins (*Van Treeck et al., 2018*). Phase separation of 'naked RNA' has been proposed to drive the assembly of stress granules, RNA granules that form under conditions of translational stress when thousands of mRNA molecules are released from polysomes (*Bounedjah et al., 2014*; *Van Treeck and Parker, 2018*). Disordered domains in proteins interact with RNA and readily phase separate with RNA in vitro (*Zagrovic et al., 2018*; *Hentze et al., 2018*), but whether these domains participate directly in mRNA recruitment in vivo has not yet been demonstrated. Here, we report that intrinsically-disordered proteins play an essential role in RNA recruitment and condensation in the context of P granules.

The P granules of *C. elegans* are a well-studied model for cytoplasmic RNA granules (*Strome, 2005*; *Seydoux, 2018*; *Marnik and Updike, 2019*). P granules are present throughout germline development; in this study, we focus exclusively on embryonic P granules. Embryonic P granules are heterogeneous assemblies (*Wang et al., 2014*) that contain at least two phases with distinct dynamics: a liquid phase assembled by the RGG domain proteins PGL-1 and its paralog PGL-3 (*Brangwynne et al., 2009*; *Updike et al., 2011*; *Hanazawa et al., 2011*; *Saha et al., 2016*), and a supporting gel-like phase, assembled by the intrinsically-disordered protein MEG-3 and its paralog MEG-4 (*Putnam et al., 2019*). MEG-3 forms small, non-dynamic condensates (<500 nanometers) that associate with the surface of larger, dynamic PGL condensates (>500 nanometers). MEG condensates enrich in the posterior cytoplasm of zygotes where they recruit and stabilize PGL condensates inherited from the oocyte (*Wang et al., 2014*; *Putnam et al., 2019*). Preferential assembly of P granules in the zygote posterior ensures their preferential inheritance by germline blastomeres (*Figure 1A*). P granules contain polyadenylated mRNAs (*Seydoux and Fire, 1994*), but so far only four mRNAs have been reported to localize to P granules in embryos (*pos-1*, *mex-1*, *gld-1* and *nos-2*; (*Subramaniam and Seydoux, 1999*; *Schisa et al., 2001*). *nos-2* codes for a homolog of the conserved germline determinant Nanos that specifies germ cell fate redundantly with *nos-1*, another Nanos homolog expressed later in development (*Subramaniam and Seydoux, 1999*). The mechanisms that recruit *nos-2* and other maternal mRNAs to P granules are not known but could involve phase separation with PGL or MEG proteins since both have been reported to phase separate with RNA in vitro (*Saha et al., 2016*; *Smith et al., 2016*).

In this study, we use immunoprecipitation, genetic and in situ hybridization experiments to characterize the P granule transcriptome. We find that mRNA recruitment to P granules occurs independently of PGL proteins and correlates directly with binding to MEG-3. MEG-3 binds ~500 mRNAs in a sequence-independent manner that favors long RNAs with low ribosome occupancy. MEG-3 condenses with RNA to form non-dynamic gel-like condensates. Our findings reveal similarities between P granules and stress granules and demonstrate a direct role for intrinsically disordered proteins in sequence-independent recruitment of mRNAs to RNA granules in vivo.

## Results

### Immunoprecipitation with MEG-3 identifies P granule mRNAs

To identify RNAs that associate with P granules in vivo, we performed individual-nucleotide resolution UV crosslinking and immunoprecipitation (iCLIP) experiments (*Huppertz et al., 2014*) on MEG-3 and PGL-1 proteins tagged at each locus with GFP. We chose these two proteins because MEG-3 and PGL-1 are essential (with their respective paralogs MEG-4 and PGL-3) to assemble the gel (MEG) and liquid (PGL) phases of embryonic P granules (*Updike et al., 2011*; *Hanazawa et al., 2011*; *Putnam et al., 2019*). Early embryos (1 to 100 cell stage) were exposed to ultraviolet light, lysed, and the cross-linked protein/RNA complexes were immunoprecipitated using an anti-GFP antibody (*Figure 1—figure supplement 1A*). As a control, we also used embryos expressing GFP alone. The GFP immunoprecipitates were washed stringently, lightly treated with nuclease to trim the bound RNAs, extracted for RNA and deep-sequenced. Sequencing reads were mapped back to the *C. elegans* genome (ws235) and used to determine read counts per locus. Read counts obtained in the control GFP iCLIP were used to define a background threshold. We identified 657 transcripts that were reproducibly recovered above the GFP background threshold in two independent MEG-3::GFP iCLIP experiments ('MEG-3-bound transcripts'; *Figure 1B*, *Figure 1—figure supplement 1B* and *Supplementary file 1*). In contrast, we identified only 18 transcripts above background in the two PGL-1::GFP iCLIPs, despite abundant PGL-1::GFP protein in the immunoprecipitates (*Figure 1B*, *Figure 1—figure supplement 1A* and *Supplementary file 1*). 15/18 of the PGL-1-bound transcripts were also in the MEG-3-bound list (*Supplementary file 1*). We compared the average normalized read count (RPKM) across the two iCLIPs to transcript abundance in P blastomeres (*Figure 1C* and *Figure 1—figure supplement 1C*) or in whole embryos (*Figure 1—figure supplement 1D*), and detected no strong correlation, suggesting that MEG-3 binds a specific subset of mRNAs. Low abundance mRNAs, however, were underrepresented in the iCLIPs and therefore could have been missed in our analysis (*Figure 1—figure supplement 1C–D*).

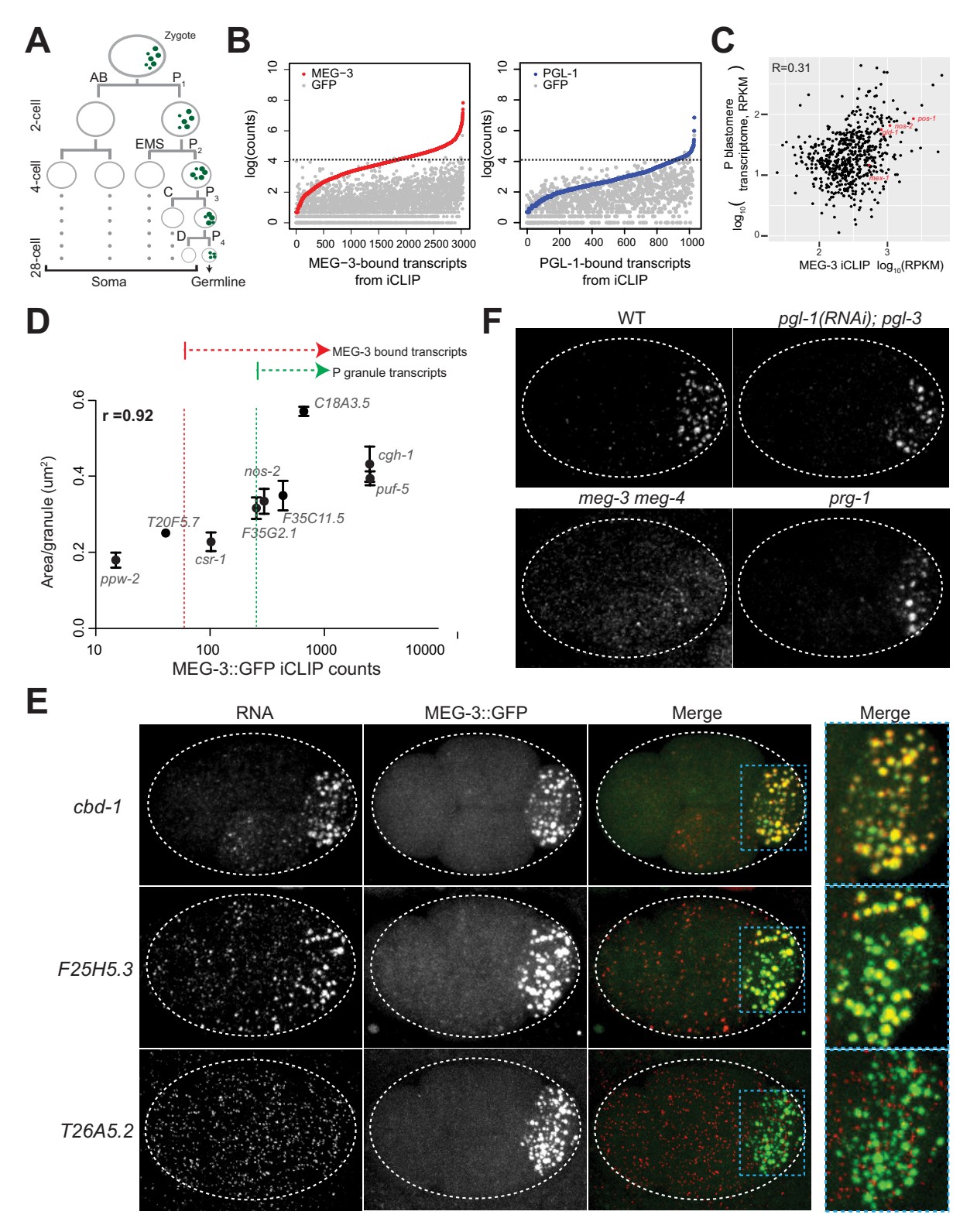

**Figure 1.** mRNAs are recruited into P granules by the MEG phase. (**A**) Abbreviated embryonic lineage showing the somatic (AB, EMS, C, and D) and germline (P) blastomeres. Green dots represent P granules. P$_4$ is the founder cell of the germline. Dotted lines refer to additional divisions not shown. (**B**) Graphs showing log transformed average read counts (Y axis) from two MEG-3::GFP (red), PGL-1::GFP (blue) and GFP (gray) iCLIP experiments. Genes are arranged along X axis based on the ascending log transformed read counts in the MEG-3::GFP or PGL-1::GFP iCLIP experiments (average of

*Figure 1 continued on next page*

*Figure 1 continued*

two experiments). Gray dots represent the GFP iCLIP read counts for each rank-ordered gene. The stippled line denotes the GFP background threshold (read counts = 60) above which transcripts were considered true positives (657 transcripts in the MEG-3::GFP iCLIPs and 18 transcripts in the PGL-1::GFP iCLIPs). (C) Graph showing the 657 MEG-3-bound transcripts (black dots) with respect to read counts in the MEG-3::GFP iCLIPs (average FPKM of two replicates, X axis) versus transcript abundance in embryonic P blastomeres (Y-axis, *Lee et al., 2017*). R is the Pearson correlation coefficient. *nos-2, mex-3, gld-1* and *pos-1* are highlighted in red. See *Figure 1—figure supplement 1* for graphs showing same for all embryonic transcripts. (D) Graph showing average read counts in MEG-3::GFP iCLIPs versus average RNA cluster size as measured from smFISH signal for nine genes. R is the Spearman correlation coefficient. Red stippled line denotes threshold for MEG-3::GFP-bound mRNAs as defined in A. Green stippled line denotes threshold for P granule mRNAs as defined in text. (E) Photomicrographs of embryos expressing MEG-3::GFP hybridized with single molecule fluorescence (smFISH) probes as indicated. *cbd-1* and *F25H5.3* are examples of transcripts localizing to P granules, as shown by colocalization with MEG-3::GFP in the right-most panels. *T26A5.2* is an example of a transcript that does not enrich in P granules. (F) Photomicrographs of 4 cell embryos of the indicated genotypes hybridized with smFISH probes against the *nos-2* transcript. All genotypes show localization of *nos-2* to P granules in the $P_2$ blastomere, except for *meg-3 meg-4*. *nos-2* is a maternal mRNA that is rapidly degraded in somatic blastomeres (two anterior cells) and therefore present at higher levels in P blastomeres and their newly born sister blastomeres (two posterior cells).

The online version of this article includes the following source data and figure supplement(s) for figure 1:

**Source data 1.** Data used to generate *Figure 1D*.
**Figure supplement 1.** mRNAs are recruited into P granules by the MEG phase.
**Figure supplement 2.** In situ hybridization.

Among the MEG-3-bound transcripts were *nos-2, pos-1, mex-1* and *gld-1*, the four transcripts previously reported to localize to P granules (*Figure 1C*). To determine whether other MEG-3-bound transcripts localize to P granules, we used single molecule fluorescent in situ hybridization (smFISH) (*Raj et al., 2008*) to examine the distribution of various transcripts in embryos. We initially analyzed nine transcripts: *nos-2*, six other transcripts in the MEG-3-bound list, and two transcripts recovered in the MEG-3 iCLIPs that did not meet the GFP background cut off. For each transcript, we determined the average granule size in the $P_2$ blastomere and compared that to the average raw read count in the MEG-3::GFP iCLIPs and observed a strong correlation (R = 0.92) (*Figure 1D*). In this analysis, *nos-2* clustered with two other genes also in the MEG-3-bound list (*F35G2.1 and F35C11.5*). Extrapolating from this correlation, we predicted that transcripts that ranked higher than the *nos-2* cluster in the MEG-3-bound list should all localize to P granules. In total, we examined 18 transcripts among this 492-gene set and found that, as predicted, all localized robustly to P granules (*Figure 1E*, *Figure 1—figure supplement 1E* and *Figure 1—figure supplement 2*). We also examined seven transcripts that ranked below the *nos-2* cluster and found none that localized to P granules in all P blastomeres, although we observed occasional, weak localization to P granules (*Figure 1—figure supplement 2*; *csr-1, R04D3.3, sip-1, ZC155.4, gly-20, T20F5.7* and *fib-1*). Finally, we examined six genes that were not recovered reproducibly in the iCLIPs and had above average expression in P blastomeres (RPKM = 7). We found none that localized to P granules (*Figure 1E* (*T26A5.2*), *Figure 1—figure supplement 2*). We conclude that ranking at or above the *nos-2* cluster in the MEG-3-bound list is a stringent metric for predicting transcripts with robust P granule localization. We designate this 492-gene set as 'P granule transcripts' (*Supplementary file 1*). This gene list corresponds to the top 75% of the 657 MEG-3-bound transcripts, and ~3% of all transcripts expressed in early embryos (15,345 transcripts detected by RNAseq) (*Lee et al., 2017*).

For all 18 P granule transcripts analyzed by smFISH, localization to P granules was inefficient with many molecules also found diffusely in the cytoplasm (*Figure 1E*, *Figure 1—figure supplement 1E*, *Figure 1—figure supplement 2*). We calculated the number of molecules in P granules and cytoplasm for two transcripts among the top five ranked in the MEG-3-bound list. We found that only 21 ± 3% *puf-5* and 34 ± 3% *Y51F10.2* molecules localized to P granules in the $P_2$ blastomere. Because P granules occupy only a small portion of the $P_2$ cell volume (5.9 ± 2%), this enrichment corresponds to a ~ 6 fold increase in concentration over the cytoplasm.

The iCLIP results suggest that mRNAs are recruited to P granules as part of the MEG phase rather than the PGL phase. Consistent with this hypothesis, *nos-2* mRNA still localized to granules in embryos lacking *pgl-1* and *pgl-3*, but not in embryos lacking *meg-3* and *meg-4* (*Figure 1F*). We obtained similar results in in situ hybridization experiments against poly-A RNA and against *Y51F10.2*, one of the transcripts identified in both the MEG-3-bound and PGL-1-bound lists (*Figure 1—figure supplement 1F*, *Supplementary file 1*). In *Drosophila*, mRNAs have been proposed

to be recruited to germ granules via interaction with piRNAs complexed with the PIWI-class Argonaute Aubergine (*Vourekas et al., 2016*). We found that *nos-2* and *Y51F10.2* still localized to P granules in embryos lacking the PIWI-class Argonaute PRG-1 (*Figure 1F* and *Figure 1—figure supplement 1F*). We conclude that mRNA recruitment to embryonic P granules depends on MEG-3 and MEG-4 and does not require PGL-1 and PGL-3 or the Argonaute PRG-1.

## MEG-3 binds RNA in a sequence-independent manner that favors low ribosome-occupancy mRNAs

The majority of reads in the MEG-3::GFP iCLIPs mapped to protein-coding mRNA transcripts (*Figure 2—figure supplement 1*). The iCLIP protocol yields short (~10–30 bp) reads that correspond to sequences cross-linked to MEG-3::GFP ('toeprints') (*Huppertz et al., 2014*). Metagene analysis of the MEG-3::GFP toeprints revealed that MEG-3 binds transcripts throughout the coding and UTR regions, with a preference for 3'UTR sequences (*Figure 2A*, *Figure 2—figure supplement 1A*). We analyzed the MEG-3 toeprints for possible motifs and found no evidence for any sequence preference (Materials and methods). MEG-3-bound transcripts tended to be longer on average than other embryonic transcripts (*Figure 2B*) and were enriched for transcripts known to be targeted by translational repressors expressed in oocytes, including OMA-1, GLD-1 and LIN-41 and CGH-1 (*Boag et al., 2008*; *Scheckel et al., 2012*; *Tsukamoto et al., 2017*) (*Figure 2—figure supplement 1B*). Ribosome profiling experiments confirmed that MEG-3-bound transcripts are on average less protected by ribosomes than other embryonic transcripts (*Figure 2C*, *Figure 2—figure supplement 2A* for P granule transcripts). To determine whether low ribosome coverage can be used to predict P granule localization, we ranked embryonic mRNAs based on ribosome occupancy. We focused this analysis on a set of mRNAs previously defined as enriched in P blastomeres (*Lee et al., 2017*) to avoid mRNAs transcribed in somatic cells which could complicate the analysis. We identified 19 mRNAs that ranked in the lowest ribosome occupancy class (ribosome occupancy <0.1; *Figure 2—figure supplement 2B*). 65% of these mRNAs were among the 'P granule transcripts' set defined above, including *cbd-1* which strongly localizes to P granules (*Figure 1E*, *Figure 1—figure supplement 2*). Among the remainder, we selected three transcripts for analysis by in situ hybridization and found that all three localized weakly to P granules. These three transcripts also exhibited a low ranking in the MEG-3::GFP iCLIPs (*gly-20, ZC155.4* and *R04D3.3*, *Figure 1—figure supplement 2*). We conclude that low ribosome occupancy is one criterion for enrichment in P granules, although efficiency of enrichment in P granules may depend on other factors.

## Translational stress enhances P granule assembly

The profiling data suggest that MEG-3 does not bind to any specific RNA sequence and simply favors 'free mRNAs' not covered by ribosomes. If so, under conditions where translation is globally inhibited, previously cytoplasmic, well translated mRNAs might be expected to re-localize to P granules. We found that a brief incubation at 30°C (15 min heat shock) was sufficient to disassemble polysomes in embryos (*Figure 2—figure supplement 2C*) (*McCormick and Penman, 1969*; *Shalgi et al., 2013*). We analyzed five non-P granule transcripts, chosen for their high ribosome occupancy under non-heat shock conditions, and remarkably found that all five accumulated in P granules after heat shock (*Figure 2D–E* and *Figure 2—figure supplement 2D*). Accumulation in P granules was observed in wild-type embryos, but not in embryos depleted of MEG proteins (*Figure 2—figure supplement 2D*). We conclude that global translational inhibition is sufficient to recruit new transcripts to P granules.

We noticed that MEG-3::GFP condensates became larger upon heat shock. In zygotes, MEG-3 molecules exist in two states: a fast-diffusing, dilute pool in the cytoplasm and slow-diffusing condensed pool in P granules (*Wu et al., 2019*). Growth of MEG-3 granules under heat shock suggested that additional MEG-3 molecules condense in P granules under conditions when translation is repressed globally. To examine this further, we measured the size of MEG-3::GFP granules in embryos treated with drugs that block translation. For these experiments, we used *mex-5 mex-6* mutant embryos, which lack embryonic polarity and assemble MEG-3 granules throughout the cytoplasm (*Smith et al., 2016*). We found that treatment with puromycin, which causes ribosomes to dissociate from transcripts, increased the size of MEG-3::GFP granules by ~4 fold, as also observed following heat-shock (*Figure 2F and G*). In contrast, treatment with cycloheximide, which stalls

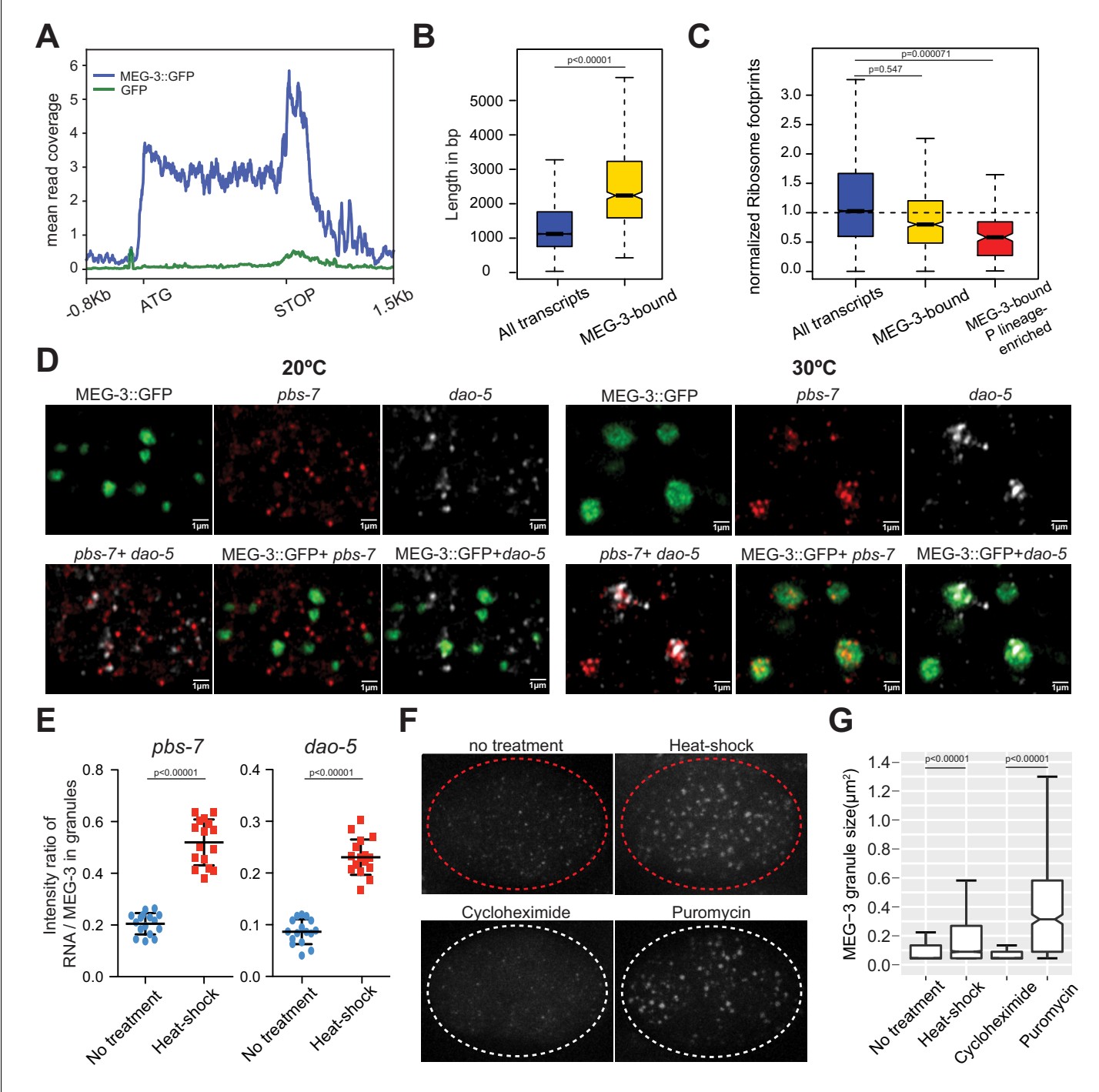

**Figure 2.** MEG proteins recruit mRNAs into P granules by a sequence non-specific mechanism that favors ribosome-poor mRNAs. (**A**) Metagene analysis of the distribution of MEG-3::GFP iCLIP reads on the 657 MEG-3-bound transcripts (blue line) compared to GFP iCLIP (green line) reads. The peak 3′ to the STOP codon correspond to 3′ UTR sequences. (**B**) Box plot showing the distribution of transcript length for embryonic transcripts (15,345 loci) versus MEG-3-bound transcripts (657 loci). P values were calculated using an unpaired t-test. Each box extends from the 25th to the 75th percentile, with the median indicated by the horizontal line; whiskers extend to the highest and lowest observations. (**C**) Box plot showing ribosome occupancy in wild-type embryos for three gene categories: embryonic transcripts (15,345 loci), MEG-3-bound transcripts (657 loci) and MEG-3-bound transcripts enriched in P lineage (187 loci). Because ribosome profiling was performed on whole embryos, footprint counts are averages across all cells. Ribosome profiles for the subset of MEG-3::GFP-bound mRNAs that are also enriched in the P lineage (*Lee et al., 2017*), therefore, are more likely representative of profiles of mRNAs in P granules. See *Figure 2—figure supplement 2A* for ribosome occupancy of P granule transcripts (MEG-3-bound RNAs above the *nos-2* cluster). P values were calculated using an unpaired t-test. Each box extends from the 25th to the 75th percentile, with

*Figure 2 continued on next page*

*Figure 2 continued*

the median indicated by the horizontal line; whiskers extend to the highest and lowest observations. (**D–E**) Photomicrographs of P$_2$ blastomeres expressing MEG-3::GFP and hybridized to probes against two non-P granule transcripts (*pbs-7* and *dao-5*) before (20˚C) and after 15 min of heat-shock (30˚C). Images are single Z sections and are representative of data quantified in (**E**). See *Figure 2—figure supplement 2D* for whole embryo images. (**E**) Graphs showing the intensity ratio of RNA over GFP in MEG-3::GFP granules under no heat-shock (blue dots) or heat-shock (red dots) conditions. Each data point represents the average value for all MEG-3::GFP granules in a single Z section (16 sections were collected from two embryos per condition). P values were calculated using an unpaired t-test. The center horizontal lines indicate the mean and bars represent the SD. (**F**) Photomicrographs of 4 cell embryos expressing MEG-3::GFP under the indicated treatments. Embryos were derived from *mex-5(RNAi) mex-6(RNAi)* hermaphrodites, which do not localize P granules (*Smith et al., 2016*). Embryos treated with cycloheximide or puromycin were derived from hermaphrodites also treated with *ptr-2(RNAi)* to permeabilize the eggshell. Images are representative of data quantified in (**G**). (**G**) Box plot showing the size distribution of MEG-3::GFP granules under different treatments as described in F. P values were calculated using an unpaired t-test. Each box extends from the 25th to the 75th percentile, with the median indicated by the horizontal line; whiskers extend to the highest and lowest observations. The online version of this article includes the following source data and figure supplement(s) for figure 2:

**Source data 1.** Data used to generate *Figure 2E*.
**Source data 2.** Data used to generate *Figure 2G*.
**Figure supplement 1.** Characterization of MEG-3 bound transcripts.
**Figure supplement 2.** MEG-3 binds ribosome-poor transcripts.
**Figure supplement 3.** Correlation analyses of sequencing libraries from wild type and *meg-3 meg4* embryos.

ribosomes on transcripts, did not change the size of MEG-3::GFP granules (*Figure 2F and G*). These findings parallel the divergent effect of puromycin and cycloheximide on the assembly of stress granules (*Kedersha et al., 2000*; *Aulas et al., 2017*). These results confirm that recruitment to P granules is driven more by translational status than by specific mRNA sequences.

## Correlation between P granule exit and translational activation for two mRNAs translated in the germline founder cell P$_4$

Among the 18 P granule transcripts analyzed by in situ hybridization, we noticed two (*nos-2* and *Y51F10.2*) that transitioned to a more diffuse cytoplasmic localization in the last P blastomere, the germline founder cell P$_4$ (*Figure 3A,B*). All other transcripts, in contrast, remained in P granules and their levels diminished starting in the P$_4$ stage as is typical for maternal mRNAs (*Figure 3—figure supplement 1A*) (*Seydoux and Fire, 1994*). As mentioned above, *nos-2* codes for a Nanos homolog that becomes translated in P$_4$ (*Subramaniam and Seydoux, 1999*). *Y51F10.2* is a new P granule transcript that had not been characterized before. We tagged the *Y51F10.2* open reading frame with a small epitope by genome editing, and found that like *nos-2*, *Y51F10.2* is translated specifically in P$_4$ (*Figure 3B*). *nos-2* translation is regulated by proteins that compete for binding to the *nos-2* 3' UTR (*Jadhav et al., 2008*). In particular, translation is repressed prior to the P$_4$ stage by the RNA-binding protein MEX-3 and activated in P$_4$ by the RNA-binding protein POS-1 (*Jadhav et al., 2008*) (*Figure 3A*). We found that the same is true for *Y51F10.2* (*Figure 3B*). In *mex-3(RNAi)* embryos, we detected NOS-2 and Y51F10.2 proteins precociously as early as the 4 cell stage. In contrast, in *pos-1 (RNAi)* embryos, NOS-2 and Y51F10.2 proteins were not expressed (*Figure 3A,B*). Remarkably, we found that these *mex-3* and *pos-1* RNAi treatments had opposite effects on RNA localization. In *mex-3(RNAi)* embryos, *nos-2* and *Y51F10.2* mRNAs did not localize to P granules. In contrast, in *pos-1(RNAi)* embryos, *nos-2* and *Y51F10.2* mRNAs remained in P granules through P$_4$ (*Figure 3A,B*). These observations confirm a link between P granule localization and translational repression.

## P granules are not required for translational repression of mRNAs in P granules

Translational repression could be a cause and/or an effect of localization to P granules. To explore this possibility, we examine the translational status of *nos-2* and *Y51F10.2* in *meg-3 meg-4* mutants which lack P granules. Surprisingly, we found that the translational timing of *nos-2* and *Y51F10.2* was unaffected in *meg-3 meg-4* mutants. *nos-2* and *Y51F10.2* were translationally silent prior to the P$_4$ stage and began translation in P$_4$ in *meg-3 meg-4* mutants as in wild-type (*Figure 3A,B*), although protein levels appeared reduced (see below). To examine the translational status of other mRNAs in *meg-3 meg-4* mutants, we repeated the ribosome profiling experiments in *meg-3 meg-4* embryos. We found that MEG-3-bound transcripts as a class maintained low ribosome occupancy in *meg-3*

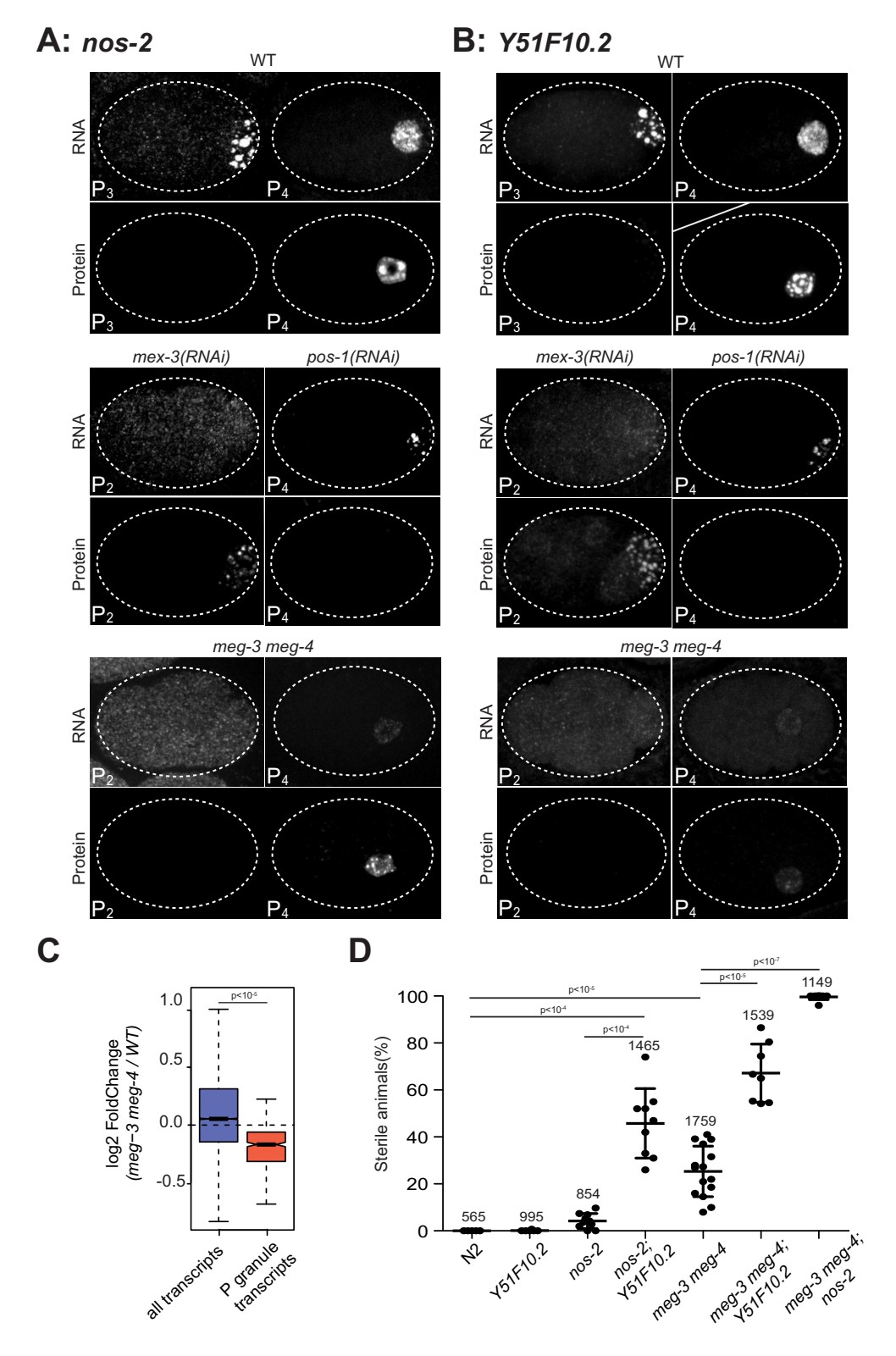

**Figure 3.** P granules enrich maternal mRNAs required for germ cell development in P blastomeres. (**A and B**) Photomicrographs of embryos of indicated stages and genotypes and hybridized to fluorescent probes or antibodies to visualize *nos-2* and *Y51F10.2* transcripts and proteins. Embryos expressing NOS-2::3xFLAG and Y51F10.2::OLLAS were used for these experiments. Note the correlation between RNA in granules and no protein expression, and RNA in the cytoplasm and protein expression in wild-type, *mex-3(RNAi)* and *pos-1(RNAi)* embryos. In *mex-3(RNAi)* embryos, *nos-2* and

*Figure 3 continued on next page*

*Figure 3 continued*

*Y51F10.2* are prematurely translated in P$_2$ where POS-1 is enriched. In *pos-1(RNAi)* embryos, *nos-2* and *Y51F10.2* are never translated. In *meg-3 meg-4* embryos, *nos-2* and *Y51F10.2* RNAs are not in P granules and thus are not preferentially segregated to P$_4$, resulting in lower RNA levels in that cell. Protein expression is correspondingly reduced but still activated at the correct stage, demonstrating that P granules are not essential for translational repression or activation. (C) Box plot showing the fold change in abundance between wild-type and *meg-3 meg-4* embryos for 15,345 embryonic transcripts and the 492 P granule transcripts. P values were calculated using an unpaired t-test. See *Figure 2B* for box plot description. The 492 P granule transcripts are present at lower levels overall in *meg-3 meg-4* embryos compared to wild-type, consistent with equal segregation to somatic blastomeres which turn over maternal mRNAs. (D) Graphs showing the percentage of sterile animals among progeny of hermaphrodites with the listed genotypes (maternal-effect sterility). Each dot represents the brood from a single hermaphrodite allowed to lay eggs for 24 hr. Total number of progeny scored across all broods is indicated for each genotype. P values were calculated using an unpaired t-test. The center horizontal lines indicate the mean and bars represent the SD.

The online version of this article includes the following figure supplement(s) for figure 3:

**Figure supplement 1.** MEG-3 and MEG-4 are not required for translational repression of 623 mRNAs in P granules.

---

*meg-4* mutants as in wild-type (*Figure 3—figure supplement 1B*). Only 25 embryonic mRNAs showed differential ribosomal occupancy in *meg-3 meg-4* embryos versus wild-type, and strikingly most showed lower ribosome occupancy (*Figure 3—figure supplement 1C*). These results confirm that neither localization to P granules nor binding to MEG-3 is a requirement for translational silencing. Translation silencing, however, appears to be a requirement for localization to P granules, with translational activation correlating with P granule exit.

## P granules enrich mRNAs coding for germ cell fate regulators in the nascent germline

We noticed that *nos-2* and *Y51F10.2* transcript and protein levels were lower in the germline founder cell P$_4$ in *meg-3 meg-4* mutants compared to wild-type (*Figure 3A,B*). Germline blastomeres are generated by a series of asymmetric divisions that give rise to somatic blastomeres and successive germline blastomeres (*Figure 1A*) (*Strome and Wood, 1982*). In wild-type embryos, P granule-associated mRNAs segregate preferentially to germline blastomeres with the P granules during each asymmetric division (*Figure 3A–B*). In contrast, in *meg-3 meg-4* mutants, P granule-associated mRNAs segregate symmetrically at each division, resulting in lower levels in germline blastomeres compared to wild-type (*Figure 3A–B*). Maternal mRNAs are degraded more rapidly in somatic blastomeres than in germline blastomeres (*Seydoux and Fire, 1994*), and this post-division asymmetry is maintained in *meg-3 meg-4* embryos (*Figure 3A–B*). As expected for equal segregation to somatic blastomeres followed by degradation, we found that P granule transcripts were present on average at lower levels overall in *meg-3 meg-4* embryos compared to wild-type as determined by RNAseq (*Figure 3C*). We conclude that recruitment into P granules serves to enrich maternal mRNAs in germline blastomeres, where mRNAs are stabilized by a P granule-independent mechanism.

30% of *meg-3 meg-4* mutants develop into sterile adults (*Figure 3D*) (*Wang et al., 2014*), raising the possibility that failure to enrich maternal mRNAs in the germline founder cell P$_4$ compromises germline development. This hypothesis predicts that *meg-3 meg-4* mutants should be hyper-sensitive to loss of P granule-associated mRNAs coding for germ cell fate determinants. To explore this prediction, we examined the effect of combining deletions in *nos-2* and *Y51F10.2* with deletions in *meg-3* and *meg-4*. Embryos derived from mothers carrying deletions in *nos-2* or *Y51F10.2* were close to 100% fertile (*Figure 3D*). In contrast, *Y51F10.2; nos-2* double mutant mothers laid 46 ± 15% sterile progeny that lacked a germline (maternal-effect sterility), suggesting that *Y51F10.2* functions redundantly with *nos-2* in germ cell fate specification (*Figure 3D*). Remarkably, the majority of *nos-2; meg-3 meg-4* and *Y51F10.2; meg-3 meg-4* triple mutants grew into sterile animals lacking a germline (*Figure 3D*). This synthetic effect is consistent with a role for *meg-3* and *meg-4* in concentrating transcripts coding for germ cell fate regulators, including *nos-2* and *Y51F10.2*, beyond a threshold required for robust germline development. The higher penetrance maternal-effect sterility of *nos-2; meg-3 meg-4* mutants compared to *Y51F10.2; nos-2* mutants suggests that other P granule transcripts also contribute to germ cell fate (*Figure 3D*). We conclude that the primary function of the MEG-3 phase is to preferentially segregate maternal mRNAs to the germline founder cell P$_4$ to ensure robust germ cell specification. Importantly, the MEG-3 phase is NOT essential for

translational silencing or for preferential stabilization of maternal mRNAs in germline blastomeres compared to somatic blastomeres.

## MEG-3 condenses with RNA to form non-dynamic nanoscale condensates

The iCLIP toeprints suggest that MEG-3 binds mRNAs with no sequence specificity. We showed previously that recombinant MEG-3 readily condenses with total RNA extracted from *C. elegans* embryos (*Putnam et al., 2019*). To investigate whether MEG-3 shows any bias when presented with specific sequences, we synthesized nine fluorescently labeled RNAs (800-1300nt size range) corresponding to embryonic transcripts with strong, minimal, or no localization to P granules in vivo under normal culture conditions. Each transcript (20 ng/μL) was combined with recombinant His-tagged MEG-3 (500 nM; *Figure 4—figure supplement 1A*) in condensation buffer containing 150 mM salt. In the absence of RNA, MEG-3 formed irregular assemblies with a broad size range (*Figure 4A*). Addition of RNA led to the formation of more uniformly sized assemblies with radii of less than 400 nm in size (*Figure 4A* and *Figure 4—figure supplement 2A,B*), consistent with the size of MEG-3 condensates in vivo (*Putnam et al., 2019*). We used this size difference to distinguish between 'aggregates' that form independent of RNA, and 'condensates' that form with RNA (*Figure 4B*; Materials and methods). We found that all nine transcripts stimulated the formation of MEG-3 condensates and became enriched in the condensates with similar efficiencies (*Figure 4B–C*, *Figure 4—figure supplement 2C*). We observed no RNA condensates in the absence of MEG-3 even at high RNA concentrations (80 ng/μL), confirming that our conditions do not induce RNA-only aggregation (*Figure 4—figure supplement 1B*).

MEG-3 has a predicted pI of 9.3 (ExPASy) and thus could potentially interact with the sugar-phosphate backbone of RNA through electrostatic interactions. To explore this hypothesis, we examined MEG-3 condensation behavior in the presence of varying concentrations of salt and RNA (*nos-2* transcript). Again, we distinguished aggregates from condensates based on size. In the absence of RNA, MEG-3 formed aggregates under all salt concentrations tested (*Figure 4D*, *Figure 4—figure supplement 2D*). In high salt (500 mM) conditions, MEG-3 continued to form aggregates even in the presence of *nos-2* RNA and these aggregates did not recruit *nos-2* RNA (*Figure 4D*, *Figure 4—figure supplement 2D,E*). Decreasing salt concentrations and increasing RNA concentrations shifted the balance from MEG-3 aggregates to MEG-3/RNA condensates. Remarkably, at the lowest concentrations of NaCl, high concentrations of RNA caused MEG-3 to solubilize with no visible aggregates or condensates (*Figure 4D*, *Figure 4—figure supplement 2D,F,G*). These observations are consistent with MEG-3 interacting with RNA in a salt-sensitive manner and suggest that electrostatic interactions with RNA compete with the MEG-3/MEG-3 interactions that lead to aggregation.

To determine whether RNA length affects MEG-3 condensation behavior, we tested RNAs of varying sizes in the condensation assay. We found that short RNAs (30 and 100 nt) were not as efficient as longer RNAs at stimulating MEG-3 condensation (*Figure 5A,B*, *Figure 5—figure supplement 1A*). We previously showed that MEG-3 protein becomes immobilized in MEG-3/RNA condensates, with no MEG-3 exchange detected within one minute of condensation (*Putnam et al., 2019*). To determine whether RNAs also become trapped in MEG-3 condensates, we performed FRAP experiments to measure the rate of RNA exchange between dilute and condensed phases comparing RNAs of different lengths. We found that short RNAs (30 and 100 nt) were mobile in MEG-3 condensates. In contrast, longer RNAs (200 nt and higher), including four full-length transcripts, exhibited no detectable exchange (*Figure 5C,D*, *Figure 5—figure supplement 1B*). We conclude that MEG-3 interacts most efficiently with mRNA-sized RNAs, which become trapped in the MEG-3 condensates.

To examine whether mRNAs also associate stably with the MEG phase of P granules in vivo, we labeled permeabilized live embryos with the RNA dye SYTO 14. As expected, we observed intense SYTO 14 fluorescence in MEG-3-positive granules (*Figure 5E*, *Figure 5—figure supplement 1C*). We verified in vitro that SYTO 14 fluorescence is sensitive to RNA and does not interact significantly with MEG-3 or PGL-3 in the absence of RNA (*Figure 5—figure supplement 1D*). When released from embryos by laser puncture of the eggshell, MEG-3 granules remain stable in aqueous buffer, whereas PGL-1 and PGL-3 dissolve immediately (*Putnam et al., 2019*). We found that MEG-3 granules remained positive for SYTO 14 ex vivo for over 1.5 min (the maximum time tested; *Figure 5E, F*). These observations indicate that mRNAs associate stably with the MEG phase ex vivo as

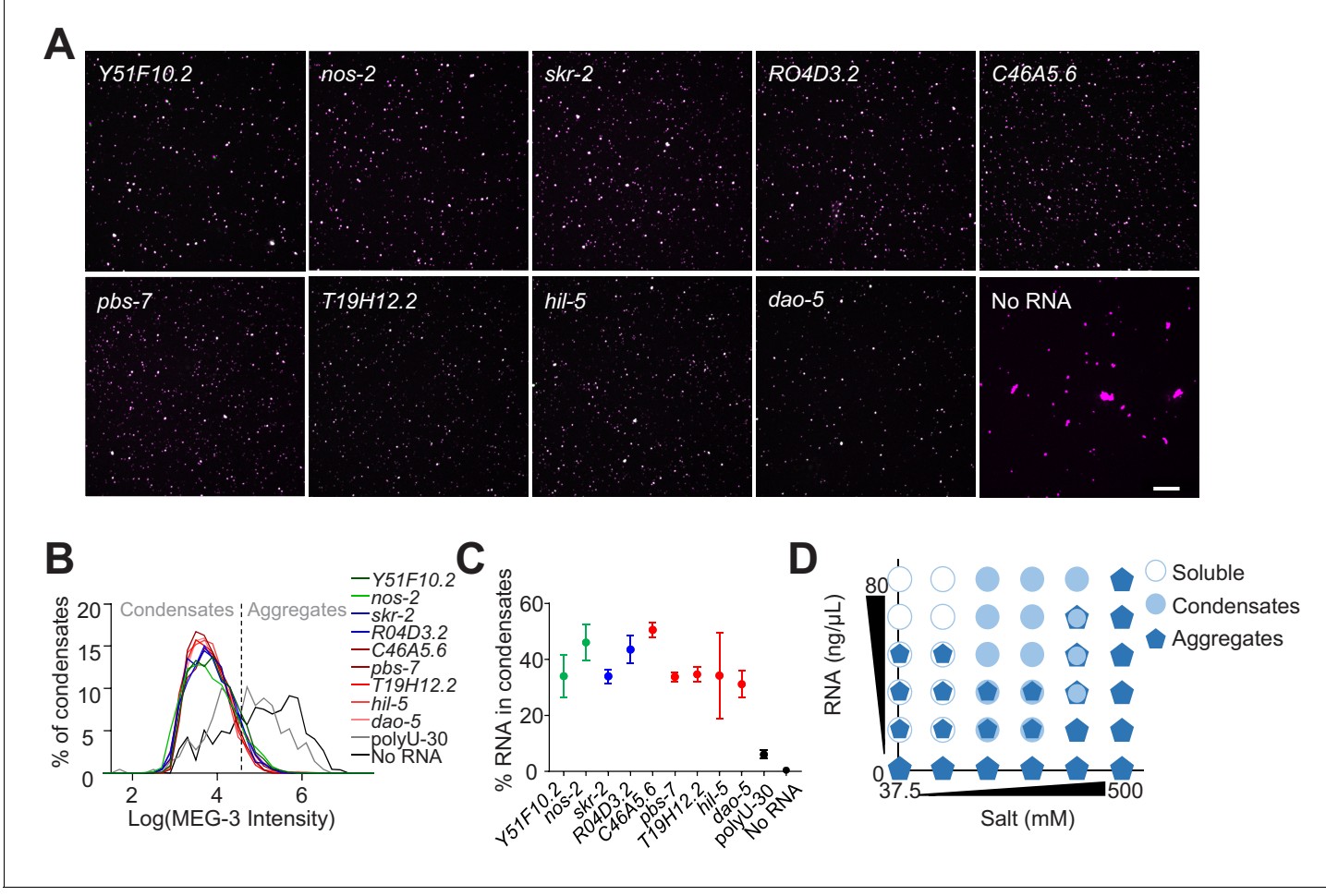

**Figure 4.** Non-sequence-specific condensation of MEG-3 with RNA. (**A**) Representative photomicrographs of condensates of MEG-3 and indicated RNA after incubation in condensation buffer. Reactions contained 500 nM MEG-3 and 20 ng/μL in vitro transcribed RNA. MEG-3 (magenta) was trace labeled with Alexa647 and RNA (green) was trace labeled with Alexa546 (Materials and methods). Scale bar is 20 μm. (**B**) Histograms of MEG-3 intensity (log10 scale) normalized to the total number of condensates in each reaction assembled as in (**A**) for each RNA indicated. Each histogram includes condensates from 12 images collected from three experimental replicates. RNAs correspond to transcripts with MEG-3 iCLIP counts above the *nos-2* cluster (*nos-2*, *Y51F10.2*), below the *nos-2* cluster (*skr-2*, *R04D3.2*) and not recovered in the MEG-3 iCLIPs (*C46A5.6*, *pbs-7*, *T19H12.2*, *hil-5*, *dao-5*). Intersection between No RNA, polyU-30 and mRNA histograms was used to quantify the fraction of MEG-3 in condensates or aggregates as indicated by dashed line. (**C**) Graph showing the percent of RNA fluorescence in MEG-3 condensates compared to total RNA assembled as in (**A**). Each data point represents condensates from 12 images collected from three experimental replicates. Circles indicate the mean and bars represent the SD. RNAs corresponding to transcripts with MEG-3 iCLIP counts above the *nos-2* cluster (green), below the *nos-2* cluster (blue) and not recovered in the MEG-3 iCLIP (red). (**D**) Phase diagram of MEG-3 condensate composition under varying RNA and salt concentrations. For representative images and quantitation corresponding to positions in the diagram refer to *Figure 5—figure supplement 1D–G*. MEG-3 was present in three states: i) soluble MEG-3 (no condensates detected, open circles), ii) small uniform condensates (Log(I)≤4.6, filled circles), iii) large irregular aggregates (Log(I)>4.6, pentagons). In conditions with mixed MEG-3 states, the larger object represents the predominant population. See *Figure 4—figure supplement 2D* for representative images.

The online version of this article includes the following source data and figure supplement(s) for figure 4:

**Source data 1.** Data used to generate *Figure 4B*.
**Source data 2.** Data used to generate *Figure 4C*.
**Figure supplement 1.** MEG-3 Purification and RNA only condensation.
**Figure supplement 2.** RNA modulates MEG condensation dependent on salt concentration.

observed in vitro, and confirm that the majority of RNAs in embryonic P granules do not reside in the PGL phase.

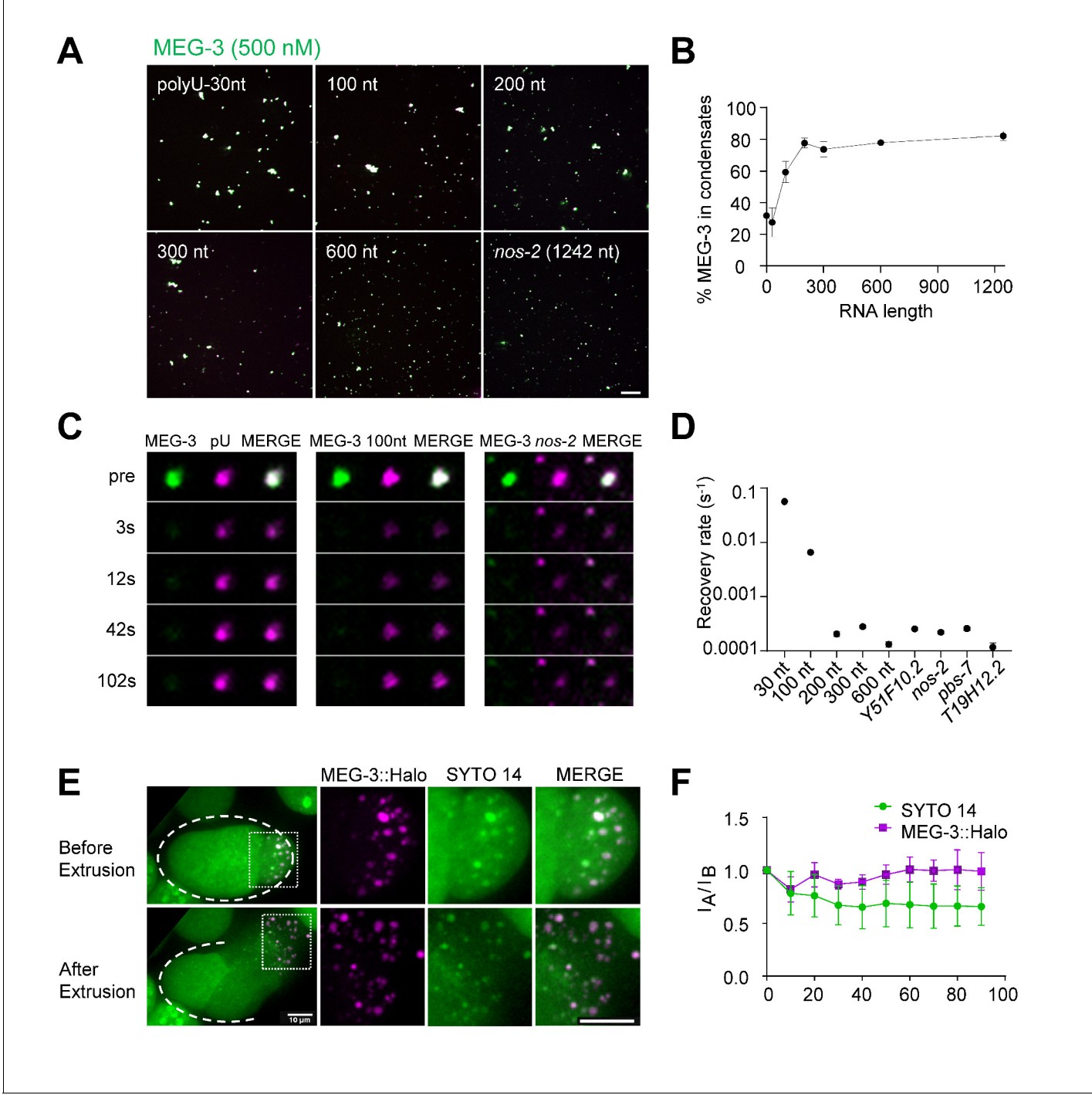

**Figure 5.** Long RNAs stably associate with the MEG gel phase. (**A**) Representative photomicrographs of MEG-3 condensation reactions with indicated RNAs. 100–600 nt RNAs are fragments of *nos-2* (Materials and methods). Reactions contained 500 nM MEG-3 and 20 ng/μL RNA, and salt. MEG-3 (green) was trace labeled with Alexa647 and *nos-2* RNAs (magenta) were trace labeled with Alexa488 or Alexa546, polyU-30nt was trace labeled with fluorescein (Materials and methods). Scale bar is 20 μm. (**B**) Percent of MEG-3 in condensates plotted vs. RNA length assembled as in (**A**). Condensates were defined as objects with a MEG-3 intensity of Log (I) ≤ 4.6 from histograms in *Figure 5—figure supplement 1A*. Each data set includes condensates from 12 images collected from three experimental replicates. Circles indicate the mean and bars represent the SD. (Materials and methods). (**C**) Representative images showing fluorescence recovery after partial photobleaching (FRAP) of condensates assembled as in (**A**) and incubated for 30 min in condensate buffer. (**D**) Graph showing rates of fluorescence recovery after photobleaching (FRAP) for indicated RNAs in MEG-3 condensates. Values were normalized to initial fluorescence intensity, corrected for photobleaching and plotted. Circles indicate the mean (n > 6) and bars represent the SD. Refer to *Figure 5—figure supplement 1B* for time traces. (**E**) Time-lapse photomicrographs of a four-cell embryo

*Figure 5 continued on next page*

*Figure 5 continued*

expressing MEG-3::Halo and stained with SYTO 14 before and 30 s after laser puncture of the eggshell. MEG-3::Halo and SYTO 14 persist in the granule phase. Scale bar is 10 µm. Quantified in *Figure 5F*. (F) Graphs showing the fraction of MEG-3::Halo or SYTO 14 retained in the condensate phase after extrusion from embryos normalized to the fraction before extrusion (time 0). Total Halo or SYTO 14 fluorescence in granules was measured before laser puncture ($I_B$) and after laser puncture ($I_A$), corrected for photobleaching and used to calculate a fluorescence ratio ($I_A/I_B$). Means are indicated along with error bars representing ± SD calculated from five embryos.

The online version of this article includes the following source data and figure supplement(s) for figure 5:

**Source data 1.** Data used to generate *Figure 5B*.
**Source data 2.** Data used to generate *Figure 5D*.
**Source data 3.** Data used to generate *Figure 5F*.
**Figure supplement 1.** RNA modulates MEG condensation dependent on RNA length.

## Discussion

### mRNAs are recruited to P granules by sequence non-specific condensation with intrinsically-disordered proteins

P granules were the first RNA granules reported to behave like 'liquid droplets' based on observations of the PGL phase (*Brangwynne et al., 2009*). In this study, we demonstrate that the PGL phase of P granules is neither necessary not sufficient for RNA recruitment to embryonic P granules, which occurs through an independent gel-like phase organized by MEG-3 (and its paralog MEG-4). Several lines of evidence indicate that mRNAs are recruited to P granules by direct binding to MEG proteins. First, iCLIP experiments demonstrate a strong correlation between MEG-3-binding and P granule localization (*Figure 1D*). Second, localization to granules requires *meg-3* and *meg-4* and does not require *pgl-1* and *pgl-3* (*Figure 1F* and *Figure 1—figure supplement 1F*). Third, mRNAs in P granules are in a phase that is resistant to dilution and high temperature like the MEG phase and unlike the PGL phase. Finally, despite lacking a canonical RNA binding domain, MEG-3 binds RNA robustly in vitro (*Smith et al., 2016*) and in vivo (*Figure 1—figure supplement 1A*), and properties of MEG-3/RNA condensates assembled in vitro match properties of the MEG phase in vivo (*Figure 4* and *Figure 5*).

We previously showed that the MEG phase is gel-like (*Putnam et al., 2019*) and we demonstrate here that mRNAs become kinetically trapped in the MEG phase. Low RNA dynamics have also been reported in the germ granules of other organisms (*Jamieson-Lucy and Mullins, 2019*; *Trcek and Lehmann, 2019*). Specification of the germline using germ granules and other asymmetrically-segregated maternal factors (germ plasm) is a derived trait that arose independently several times during metazoan evolution (*Kulkarni and Extavour, 2017*). Proteins that scaffold germ granule assembly in different organisms *Drosophila*Oskar, vertebrate Xvelo/Bucky Ball, and *C. elegans* MEGs) are non-homologous, but all contain predicted intrinsically-disordered domains and form non-dynamic condensates in vitro. Oskar and Bucky Ball bind Nanos RNA in vitro (*Yang et al., 2015*; *Krishnakumar et al., 2018*), as we show here for MEG-3. Condensation of disordered protein domains with RNA, therefore, may be a common driver of germ granule assembly in a wide range of animals. Trapping RNAs in a solid, gel-like phase could be beneficial for long term storage of RNAs during oogenesis and for maintaining a pool of maternal mRNAs in germline precursors before the onset of zygotic transcription ($P_4$ stage; *Seydoux and Dunn, 1997*).

What defines which mRNAs are recruited into germ granules? We report here that recruitment of *nos-2* RNA to P granules requires MEX-3, a translational repressor that recognizes specific sequences in the *nos-2* 3' UTR (*Jadhav et al., 2008*). Similarly, in vertebrate embryos, mRNA recruitment to the Balbiani body depends on 3' UTR sequences recognized by sequence-specific mRNA-binding proteins (*Jamieson-Lucy and Mullins, 2019*). In *Drosophila*, recruitment to polar granules also depends on 3'UTR sequences implicated in translational repression, as well as potential RNA:piRNA and RNA:RNA homotopic interactions (*Trcek and Lehmann, 2019*). We suggest that sequence-specific protein:RNA and/or RNA:RNA interactions involving 3' UTR sequences define a pool of low translation mRNAs. By virtue of its low ribosome occupancy, this pool is preferentially captured by non-sequence specific, intrinsically-disordered proteins in germ plasm for condensation and segregation to the embryonic germline.

## Parallels between embryonic P granules and stress granules

Our findings highlight several similarities between embryonic P granules and stress granules. Stress granules are RNA granules that form in the cytoplasm of stressed cells to store mRNAs that have exited the translational pool (*Buchan and Parker, 2009*; *Ivanov et al., 2019*). As we report here for P granules, the stress granule transcriptome favors long mRNAs that are ribosome-depleted (*Kedersha et al., 2000*; *Khong et al., 2017*; *Aulas et al., 2017*). Stress granule assembly is enhanced by treatments that release mRNAs from polysomes (heat-shock and puromycin), as we show here for P granules. Recruitment of mRNAs to stress granules is an inefficient process, with only a minority of molecules for most mRNA species localizing in the granules (*Khong et al., 2017*). Recruitment into P granules is also inefficient with only ~30% of molecules localizing to granules for two of the most robust P granule transcripts described here. Translational repression is required for localization to P granules, but P granules are not required for translational repression or for mRNA stability, as is also true for stress granules (*Kedersha et al., 2016*). Finally, P granules contain proteins also found in stress granules (poly-A binding protein, TIA-1 and the DDX3/LAF-1 RNA helicase) and P granules interact with P bodies, as do stress granules (*Gallo et al., 2008*; *Shih et al., 2012*; *Elbaum-Garfinkle et al., 2015*). Our findings suggest that, like stress granules, embryonic P granules function downstream of the translational regulation machinery to temporarily hold mRNAs not engaged with ribosomes until their degradation or translation in the germline blastomere $P_4$. A recent pre-print (*Parker et al., 2020*) also reports that translational repression is a prerequisite for RNA localization to P granules. Interestingly, P bodies also have been reported to favor translationally repressed mRNAs and to recruit additional mRNAs under translational stress (*Hubstenberger et al., 2017*; *Matheny et al., 2019*). Low translation, therefore, may be a common requirement for entry into cytoplasmic granules. Our analyses indicate, however, that not all translationally-repressed mRNAs are recruited equally strongly to P granules. A key question for the future will be to understand what factors beyond low translation contribute to enrichment in P granules.

## P granules enrich maternal mRNAs in the germline founder cell $P_4$ to maximize robustness of germ cell fate specification

What is the function of P granules? A unique characteristic of P granules is their polarized assembly (*Strome and Wood, 1982*). P granules assemble preferentially in cytoplasm destined for germline blastomeres (*Smith et al., 2016*) and consequently mRNAs in P granules are preferentially segregated to germline blastomeres during the first embryonic cleavages. Because mRNA localization to P granules is inefficient, this enrichment is relatively weak but, over 4 cell divisions, boosts mRNAs levels in the germline founder cell $P_4$ over what would have been achieved by equal segregation. *meg-3 meg-4* mutants, which lack P granules, segregate P granule mRNAs equally to germline and somatic blastomeres, leading to reduced levels in $P_4$ compared to wild-type. *meg-3 meg-4* mutants are 30% sterile and exquisitely sensitive to loss of germ cell fate regulators. These observations are consistent with previous studies, which showed that embryonic P granules, while non-essential, are required for robust germ cell fate specification (*Gallo et al., 2010*). Embryonic P granules are also required to maintain small RNA homeostasis across generations, likely via the PGL phase which contains Argonaute proteins and other epigenetic factors (*Ouyang et al., 2019*; *Dodson and Kennedy, 2019*). We propose that embryonic P granules are two-phase, dual-cargo condensates whose main function is to maximize transmission of maternal mRNAs (MEG phase) and epigenetic factors (PGL phase) to the germline founder cell $P_4$. Importantly, our findings demonstrate that P granules are not essential for translational repression or preferential RNA stability in germline blastomeres.

What P granule mRNAs code for germ cell fate regulators? Our analyses identified 492 predicted 'P granule transcripts'. This list is unlikely to be exhaustive, especially since low abundance mRNAs were not recovered efficiently in the MEG-3 iCLIP experiments. We surveyed 18 P granule transcripts by in situ hybridization and identified two that exit P granules and become translated in the germline founder cell $P_4$: the known germ cell fate regulator *nos-2* and a previously uncharacterized transcript *Y51F10.2*. *Y51F10.2* codes for the *C. elegans* orthologue of human TRIM32, a member of the broadly conserved TRIM-NHL protein family implicated in protein ubiquitination and mRNA regulation (*Tocchini and Ciosk, 2015*). Our genetic findings indicate that 1) *Y51F10.2* functions in germ cell fate specification redundantly with *nos-2* and 2) additional germ cell fate regulators likely exist among the other P granule mRNAs. It appears unlikely, however, that all P granule-enriched

transcripts function in germ cell specification. Unlike *nos-2* and *Y51F10.2*, the other P granule mRNAs we surveyed by in situ hybridization turn over before ever releasing from P granules (*Figure 3—figure supplement 1*), and at least some are known to function in unrelated processes (e.g. *cbd-1*; a chitin-binding protein required for egg shell biogenesis *Johnston et al., 2010*). The MEG phase, therefore, may serve as a relatively non-specific repository for untranslated or low-translation mRNAs, many of which do not function in germ cell fate specification. It is interesting to note that the first step for mRNA selection in the *Drosophila* polar granules also has been proposed to rely on a relatively non-specific mechanism, in that case involving mRNA:piRNA interactions (*Vourekas et al., 2016*). Similarly, recruitment of mRNAs to stress granules is mostly non-specific with over 80% of mRNA species recruited (*Khong et al., 2017*). Sequence non-specific RNA condensation may therefore be a common strategy to spatially segregate mRNAs in cells and embryos.

# Materials and methods

## Key resources table

| Reagent type (species) or resource | Designation | Source or reference | Identifiers | Additional information |
|---|---|---|---|---|
| Strain, strain background *C. elegans* | JH3503 | *Smith et al., 2016* | | *meg-3(ax3054[meg-3::meGFP]) X.* |
| Strain, strain background *C. elegans* | JH3269 | *Putnam et al., 2019* | | *pgl-1(ax3122[pgl-1::GFP]) IV.* |
| Strain, strain background *C. elegans* | JH3193 | *Paix et al., 2014* | | *nos-2(ax2049[3xFLAG::nos-2]) II.* |
| Strain, strain background *C. elegans* | JH3605 | This study | | *Y51F10.2(ax4319[Y51F10.2::OLLAS]) I* |
| Strain, strain background *C. elegans* | EGD364 | *Wu et al., 2019* | | *meg-3(egx4[meg-3::Halo]) X.* |
| Strain, strain background *C. elegans* | JH3475 | *Smith et al., 2016* | | *meg-3(ax3055) meg-4(ax3052) X* |
| Strain, strain background *C. elegans* | WM527 | *Shen et al., 2018* | | *prg-1(ne4523 [gfp::tev::flag::prg-1]) I* |
| Strain, strain background *C. elegans* | JH3357 | *Lee et al., 2017* | | *nos-2(ax3103[nos-2△]) II* |
| Strain, strain background *C. elegans* | SS608 | *Kawasaki et al., 2004* | | *pgl-3(bn103[pgl-3△]) V* |
| Strain, strain background *C. elegans* | SX922 | Caenorhabditis Genetics Center | | *prg-1(n4357[prg-1△])I* |
| Strain, strain background *C. elegans* | JH3229 | *Wang et al., 2014* | | *meg-1(vr10) meg-3(tm4259)X* |
| Strain, strain background *C. elegans* | JH3740 | This study | | *meg-3(ax3055) meg-4(ax3052) X; Y51F10.2(ok1610) I* |
| Strain, strain background *C. elegans* | JH3743 | This study | | *nos-2(ax3130) II; Y51F10.2(ok1610) I* |
| Strain, strain background *C. elegans* | JH3746 | This study | | *meg-3(ax3055) meg-4(ax3052) X; nos-2(ax3130) II. 100% sterile, no clone was maintained.* |
| Strain, strain background *C. elegans* | JH1904 | This study | | *Unc-119(ed3) III; axls1374[Ppie1::GFP]* |
| Strain, strain background *C. elegans* | JH2878 | *Leacock and Reinke, 2008* | | *meg-1(vr10) X* |
| Strain, strain background *C. elegans* | JH3562 | This study | | meg-3(ax3054[MEG-3::meGFP]) X; K08F4.2 (ax5000[*gtbp-1::tagRFP*]) IV |
| Strain, strain background *C. elegans* | RB1413 | Caenorhabditis Genetics Center | | *Y51F10.2(ok161) I* |

*Continued on next page*

*Continued*

| Reagent type (species) or resource | Designation | Source or reference | Identifiers | Additional information |
|---|---|---|---|---|
| Antibody | K76 | DSHB, PMID: 28787592 | RRID:AB_531836 | (1:15) |
| Antibody | Anti-FLAG M2 | Sigma-Aldrich Cat# F3165 | RRID:AB_259529 | (1:200) |
| Antibody | Donky-anti-mouse IgM 647 | Jackson Immuno Research Labs | RRID:AB_2340861 | (1:400) |
| Antibody | Goat anti-Rabit IgG (H+L) 568 | Molecular probes cat# A-11011 | RRID:AB_143157 | (1:400) |
| Antibody | Anti-OLLAS-L2 | Novus cat# NBP1-06713 | RRID:AB_1625979 | (1:200) |
| Antibody | Anti-OLLAS | other | | gift from Dr. Jeremy Nathans |
| Antibody | Anti-GFP | Rohe | RRID:AB_390913 | For conjugation |
| Sequence-based reagent | oCYL1089: crRNA to cut Y51F10.2 at 3' end | This study | | GTGCTCAAAATAGTAGGCGA |
| Sequence-based reagent | oCYL1143: repair oligo of Y51F10.2 C-ter Ollas tag (+) | This study | | TCCAGCGCCAGCACCACCATTCGAC AACTCCGTCGCCTACTATTTTGGAGGAT CCGGAtccggattcgccaacGAGCTCggac cacgtctcatgggaaagGGAGGATCCGG AGAGCACCAATTTTGA gcttttatatttttttttctc |
| Sequence-based reagent | oCYL1144: repair oligo of Y51F10.2 C-ter Ollas tag (-) | This study | | gagaaaaaaaaatataaaagc TCAAAATTGGTGCTCTCCGGATCCTC Cctttcccatgagacgtggtcc GAGCTCgtt ggcgaatccggaTCCGG ATCCTCCAAAAT AGTAGGCGACGGAGTTGTCGA ATGGTGGTGCTGGCGCTGGA |
| Sequence-based reagent | oCYL1096:5' PCR primer 333 bp up of Y51F10.2 TGA stop | This study | | GTTTCCAGCCGCTTGACAAG |
| Sequence-based reagent | GTTTCCAGCCGCTTGACAAG | This study | | CTGATCCTCCCCCTTCTTCG |
| Sequence-based reagent | oCYL1259:5' PCR primer contains T7 promoter for in vitro transcription of T19H12.2 mRNA. | This study | | CATGATTACTAATACG ACTCACTATA GGGaccagctcacga aactaacaatg |
| Sequence-based reagent | oCYL1260:3' PCR primer at the end of T19H12.2 3UTR for T7 in vitro transcription | This study | | gaaagcgaaagaaatttt attttacaggagg |
| Sequence-based reagent | oCYL1261:5' PCR primer contains T7 promoter for in vitro transcription of dao-5 mRNA including utr. | This study | | catgattacTAATACGACT CACTATAGGG ggtacccctgatcgctATGAG |
| Sequence-based reagent | oCYL1262:3' PCR primer at the end of dao-5 3UTR for T7 in vitro transcription | This study | | ggaccaaacattttatggat gagacaag |
| Sequence-based reagent | oCYL1263:5' PCR primer contains T7 promoter for in vitro transcription of hil-5 mRNA. | This study | | catgattacTAATACGACT CACTATAGGG actatcacttttcaagtgtttgttcatcg |
| Sequence-based reagent | oCYL1264:3' PCR primer at the end of hil-5 3UTR for T7 in vitro transcription | This study | | agaatctattaatggtttattggaa ggtatatttgttaaaatg |

*Continued on next page*

*Continued*

| Reagent type (species) or resource | Designation | Source or reference | Identifiers | Additional information |
|---|---|---|---|---|
| Sequence-based reagent | oCYL1265:5' PCR primer contains T7 promoter for in vitro transcription of pbs-7 mRNA including utr. | This study | | catgattacTAATACGACT CACTATAGGG gcatttcattgtcgaaattcacttcctttc |
| Sequence-based reagent | oCYL1266:3' PCR primer at the end of pbs-7 3UTR for T7 in vitro transcription | This study | | agaaggattaaatggaag tttatttatcgacttc |
| Sequence-based reagent | oCYL1267:5' PCR primer contains T7 promoter for in vitro transcription of T07C4.3a mRNA including utr. | This study | | catgattacTAATACGACT CACTATAGGG gtttgtgcactcactacgaaatctc |
| Sequence-based reagent | oCYL1268:3' PCR primer at the end of T07C4.3a 3UTR for T7 in vitro transcription | This study | | catcaaaatattctttcatt taacaaaaacagaaacaac |
| Recombinant DNA reagent | plasmid: 6XHis-MEG-3 | *Smith et al., 2016* | | |
| Recombinant DNA reagent | plasmid: MBP-HIS-TEV-PGL-3 | *Putnam et al., 2019* | | |
| Chemical compound, drug | SYTO 14 | ThermoFisher Cat#S7572 | | In vivo RNA labeling |
| Chemical compound, drug | Alexa Fluor 647 NHS Ester | ThermoFisher Cat#A37573 | | protein labeling |
| Chemical compound, drug | DyLight 488 NHS Ester | ThermoFisher Cat#46403 | | protein labeling |
| Chemical compound, drug | ChromaTide Alexa Fluor 488–5-UTP | ThermoFisher Cat#C11403 | | RNA labeling |
| Chemical compound, drug | ChromaTide Alexa Fluor 546–14-UTP | ThermoFisher Cat#C11404 | | RNA labeling |
| Recombinant DNA reagent | plasmid: cDNA of pbs-7 | this paper | | *pbs-7* cDNA, pUC19 vector |
| Recombinant DNA reagent | plasmid: cDNA of dao-5 | this paper | | *dao-5* cDNA, pUC19 vector |
| Recombinant DNA reagent | plasmid: cDNA of T19H12.2 | this paper | | T19H12.2 cDNA, pUC19 vector |
| Recombinant DNA reagent | plasmid: cDNA of hil-5 | this paper | | *hil-5*, pUC19 vector |
| Recombinant DNA reagent | plasmid: cDNA of Y51F10.2 | this paper | | Y51F10.2 cDNA, PCR blunt II topo vector |
| Recombinant DNA reagent | plasmid: cDNA of nos-2 | this paper | | *nos-2* cDNA, PCR blunt II topo vector |
| Recombinant DNA reagent | plasmid: cDNA of skr-2 | this paper | | *skr-2* cDNA, PCR blunt II topo vector |
| Recombinant DNA reagent | plasmid: cDNA of R04D3.2 | this paper | | R04D3.2 cDNA, PCR blunt II topo vector |
| Recombinant DNA reagent | plasmid: cDNA of C46A5.6 | this paper | | C46A5.6 cDNA, PCR blunt II topo vector |
| Software, algorithm | DESeq2 | https://bioconductor.org/ packages/release/bioc/ html/DESeq2.html | RRID:SCR_015687 | |
| Software, algorithm | hisat2 | DOI: 10.1038/nprot.2016.095 | RRID:SCR_015530 | |
| Software, algorithm | htseq-count | DOI: 10.1093/ bioinformatics/btu638 | RRID:SCR_011867 | |

*Continued on next page*

*Continued*

| Reagent type (species) or resource | Designation | Source or reference | Identifiers | Additional information |
|---|---|---|---|---|
| Software, algorithm | cuffdiff | http://cole-trapnell-lab.github.io/cufflinks/ | RRID:SCR_001647 | |
| Software, algorithm | Slidebook 6 | https://www.intelligent-imaging.com/slidebook | RRID:SCR_014300 | |
| Software, algorithm | Deeptools | https://deeptools.readthedocs.io/en/develop/ | RRID:SCR_016366 | |
| Software, algorithm | icount | https://github.com/tomazc/iCount | RRID:SCR_016712 | |
| Software, algorithm | smatools | http://samtools.sourceforge.net/ | RRID:SCR_002105 | |
| Software, algorithm | BEDTools | https://github.com/arq5x/bedtools2 | RRID:SCR_006646 | |
| Software, algorithm | Galaxy | https://usegalaxy.eu/ | RRID:SCR_006281 | |
| Software, algorithm | Rstusio | http://www.rstudio.com/ | RRID:SCR_000432 | |
| Software, algorithm | STAR | https://github.com/alexdobin/STAR | RRID:SCR_015899 | |

## Worm handling, RNAi, sterility counts

*C. elegans* were cultured according to standard methods (*Brenner, 1974*). RNAi knockdown experiments were performed by feeding on HT115 bacteria (*Timmons and Fire, 1998*). Feeding constructs were obtained from Ahringer or OpenBiosystem libraries. The empty pL4440 vector was used as negative control. Bacteria were grown at 37˚C in LB + ampicillin (100 µg/mL) media for 5–6 hr, induced with 5 mM IPTG for 30 min, plated on NNGM (nematode nutritional growth media) + ampicillin (100 µg/mL) + IPTG (1 mM) plates, and grown overnight at room temperature. Embryos isolated by bleaching gravid hermaphrodites were put onto RNAi plates directly. To culture larger number of worms for iCLIP and ribosome profiling experiments, worm cultures were started from synchronized L1s (hatched from embryos incubated in M9 overnight) onto NA22 or RNAi bacteria containing plates and grown to gravid adults. Early embryos were harvested from gravid adults.

To verify the efficiency of RNAi treatments for knocking down *meg* genes, we scored animals exposed to the same RNAi feeding conditions for maternal-effect sterility. For *meg-1(vr10)* strain on *meg-2* RNAi, sterility was 95 ± 0.6% at 20˚C; *meg-1 meg-3 meg-2(RNAi) meg-4(RNAi)* maternal effect sterility was 100 ± 0%. To verify the RNAi efficiency of targeting *mex-3* and *pos-1*, embryonic lethality was assayed. Cohorts of 10–20 mothers were allowed to lay eggs for periods ranging from 2 to 4 hr. Embryos were then counted, and adults were scored four days later. The embryonic lethality for both *mex-3* and *pos-1* were 100 ± 0% and 98 ± 0.2% respectively.

## Strain construction by CRISPR-mediated genome editing

CRISPR generated lines were created as in *Paix et al. (2017)* as indicated in the strain in the key resources table. Guides and repair temples used for CRISPR are listed in the key resources table.

## iCLIP and library preparation

### Protease inhibitor mix for immunoprecipitation

We prepared freshly-made 100x protease inhibitor mixes for the immunoprecipitation step in the iCLIP protocol. Protease inhibitor mix1(100x): Antipain (3 mg), Leupeptin(5 mg), Benzamidine(10 mg), AEBSF(25 mg) and phosphoramidon (1 mg) were resuspended in 1 mL Phosphate-buffered saline (PBS). Protease inhibitor mix2 (100x): 500 µL of 10 mg/mL Aprotinin, 400 µL of 10 mM Besttatin, 100 µL of 10 mM E64, and 100 µL of 10 mg/mL Trypsin inhibitor were mixed together in $H_2O$.

### Crosslinking, immunoprecipitation and nuclease treatments

The iCLIP protocol was adapted from *Huppertz et al. (2014)* with some modifications as detailed below. *C. elegans* embryos collected from ~$6.7 \times 10^7$ gravid adults were seeded on 10 cm petri dishes and irradiated 3 times with 400 mJ/cm$^2$ at UV 254 nm by Stratalinker. Crosslinked embryos

were collected and resuspended in Immunoprecipitation buffer (IP buffer) with freshly made protease inhibitors as described in this section [300 mM KCl, 50 mM HEPES pH7.4, 1 mM EGTA, 1 mM MgCl$_2$, 10% glycerol, 0.1% NP-40]. Samples were lysed in a Spex 6870 freezer mill followed by centrifugation at 4°C 21,000xg for 30 min to remove embryo debris. Cleared lysates were subjected to immunoprecipitation using anti-GFP antibody (Roche 11814460001) conjugated to protein G magnetic beads (Thermofisher) in the presence of 200unit of RNaseOut per milliliter of lysates. Immunoprecipitated fractions were then washed extensively with IP buffer, low salt buffer [150 mM NaCl, 50 mM HEPES pH7.4, 0.1% SDS, 0.5% NP-40, 1 mM MgCl$_2$, 1 mM EGTA] and high salt buffer [500 mM NaCl, 50 mM HEPES pH7.4, 0.1% SDS, 0.5% NP-40, 1 mM MgCl$_2$, 1 mM EGTA]. Bound fractions were then washed with 1xPBS to remove excess salt followed by RQ1 DNase treatment at 37° C for 10 min [4 units of RQ1 (Promega M6101) and 60 units of SUPERaseIN (ThermoFisher AM2694)]. To perform partial RNA digestion, RQ1 DNase treated bound fractions were washed with 1 mL low salt buffer, 1 mL high salt buffer, 1 mL 1x PNK buffer [50 mM Tris-HCl pH7.4, 10 mM MgCl$_2$, 0.5% NP-40] and finally resuspend in 500 μL MNase reaction buffer [NEB, M0247, one unit/mL in 1x reaction solution]. The MNase reaction was immediately transferred to a thermomixer for 2 min at 37°C. The MNase reaction was stopped with ice-cold 1x PNK buffer with 5 mM EGTA followed by 2 × 1 mL high salt buffer, 2 × 1 mL low salt buffer and 1 × 1 mL PNK buffer.

### L3 adapter ligation

A dephosphorylation step is necessary to remove 3′ end phosphates that prevent adapter ligation. Beads were resuspended in 20 μL of dephosphorylation reaction [4 μL of 5x PNK reaction buffer, 0.5 μL T4 PNK (NEB M0201), 0.5 μL RNasin (Promega N2111), 15 μL H$_2$O] at 37° C for 20 min followed by washes with PNK buffer. Beads were resuspended in in 20 μL ligation mixture [2 μL of 10x RNA ligation buffer, 2 μL of T4 RNA ligase II truncated KQ (NEB M0373), 0.5 μL of SUPERaseIN, 1.5 μL of 20 μM pre-adenylated L3-App adapter, 4 μL of PEG8000, 10 μL H$_2$O] at 16° C for overnight.

### 5′ end labeling, SDS-PAGE and nitrocellulose transfer

Ligated RNA samples were washed twice with high salt buffer and twice with PNK buffer. Supernatants were removed and samples were resuspended in 80 μL of hot PNK mix for 5′ end labeling [8 μL of 10x PNK buffer, 1 μL of P$^{32}$ ATP, 4 μL of T4 PNK, 67 μL H$_2$O] at 37° C for 10 min. Remove unlabeled hot ATP by wash beads with wash buffer [20 mM Tris-HCl pH7.4, 10 mM MgCl$_2$, 0.2% Tween-20]. Samples were loaded on a 4–20% TGX protein gel (Bio-Rad 4561093) and transferred to a nitrocellulose membrane.

### RNA isolation, Reverse transcription, cDNA circularization and PCR amplification

These procedures were performed as described in *Huppertz et al. (2014)*.

## Preparation of libraries for ribosome profiling

Synchronized L1 worms were seeded on plates containing HT115 bacteria transformed with pL4440 vector and cultured at 25°C for ~48 hr. Additional bacteria were added to ensure enough food to support development. Early embryos were collected by bleaching gravid hermaphrodites. Small aliquots of embryos were collected from each experiment and staged by DAPI-staining. 70 ± 7% of embryos were before or at ~100 cell stage. ~400 μL packed embryos were then resuspended in 2 mL footprint lysis buffer [20 mM Tris-Cl (pH8.0), 140 mM KCl, 1.5 mM MgCl2, 1% Triton X-100, 0.1 mg/mL CHX] and lysed in a Spex 6870 freezer mill. After clarification lysate by sequential centrifugation at 3000 rpm followed by 17,000xg, 100 μL (5% of lysate) of lysates were saved for mRNAseq. For ribosome profiling, lysates containing 300 μg of total RNA were treated with 100 units of RNaseI (Ambion) for 30 min at 25°C. 40 units of SUPERaseIN (ThermoFisher) were added to prevent further digestion from RNaseI. Monosomes were isolated by sucrose gradients (10–50%) and centrifuged at 40,000 rpm for 3 hr in a SW41 rotor (Beckman Coulter). The extracted RNA was size-selected (15–34 nt) after running on 15% denaturing PAGE gels. An oligonucleotide adapter was ligated to the 3′ end of isolated fragments. After ribosomal RNA depletion using RiboZero (Illumina), the following steps were performed: reverse transcription using SuperScript III reverse transcriptase (Thermo Fisher Scientific), circularization using CirLigase I (Lugicen) and PCR amplification (*Schuller et al.,*

*2017*). Libraries were sequenced on a HiSeq2500 machine at facilities at the Johns Hopkins Institute of Genetic Medicine.

## High-throughput sequencing analysis

### iCLIP data analysis

The 5' barcodes (NNN- four nt indexes – NN) and 3' adaptor (AGATCGGAAG) for iCLIP library construction were listed in *Huppertz et al. (2014)*. iCLIP sequencing reads were trimmed to remove 3' adaptor and 5' randomized barcodes using -fastx_clipper and custom python codes base on Ule lab GitHub depository (https://github.com/jernejule/non-coinciding_cDNA_starts). Trimmed reads were aligned to *C. elegans* ws235 reference genome using HISAT2 (*Kim et al., 2015*), and PCR duplicated reads were removed. Description of these steps and modified codes were deposited in GitHub https://github.com/fishhead1978/iCLIP_2019 (*Lee and Lu, 2020*; copy archived at https://github.com/elifesciences-publications/iCLIP_2019). To determine the distribution of mapped read across the genome, an R package RNA centric annotation system (RCAS) was used to generate the plot shown in *Figure 2—figure supplement 1A*. Reads aligning to genetic features were then counted using HTseq-count (*Supplementary file 1* and *Supplementary file 6*). This information was used to plot *Figure 1B,C*, and *Figure 1—figure supplement 1B-D*. The list of 657 'MEG-3 bound transcripts' was generated by collecting all the genes identified in both MEG-3::GFP iCLIP experiments and removing genes with read counts lower than 60 (background level based on GFP iCLIP results shown in *Figure 1B*).

To identify potential MEG-3 binding motifs, MEG-3::GFP and control GFP iCLIP mapped reads were used in peak caller PEAKachu in galaxy server (https://usegalaxy.eu/) with options –Minimum cluster Expression Fraction 0.01 –Minimum Block Overlap 0.5 –Minimum Block Expression 0.1 –Mad Multiplier 2.0 –Fold Change Threshold 1.5 – Adjusted p-value Threshold 0.1. Identified peaks with additional 15 nt extensions from both ends were used by MEME suite to search for sequence motif, and no motif reached the E-value cut off (E- value)$<1\times10{-5}$. An additional iCount analysis package (*Curk, 2019* https://icount.readthedocs.io/en/latest/) with the same options described in the tutorial document (https://icount.readthedocs.io/en/latest/ref_CLI.html) was used to identify significant peaks. 15 nt extensions were added to both sides of peaks and followed by MEME suite motif analysis. No motif with an E-value $<1\times10\text{-}5$ were found in either the MEG-3::GFP or GFP iCLIP experiments. Therefore, we conclude that MEG-3 binds RNAs without any sequence bias. The preference for 3'UTR sequences observed in the metagene analysis shown in *Figure 2A* may reflect the fact that 3'UTRs are ribosome-free.

To plot the MEG-3 binding profile (*Figure 2A*), bamCompare and computeMatrix in the deep-Tools package (http://deeptools.readthedocs.io/en/latest/) were used to compute mapped read coverage. Command lines were listed as below:

```
$ bamCoverage -b < inpit >  o<output.bw> -bs 1 p 8 -ignore chrM -exactScaling -
smoothLength 3
$ computeMatrix scale-regions -S < input.bw> -R < MEG-3-bound transcripts.bed> -b
800 -a 1500 m 2000 -bs 1 -skipZeros -sortUsing max -p 8 -o < output.gz>
$ plotProfile -m < input.gz> -out<output.pdf> -colors blue green -perGroup
```

### mRNA sequencing

Sequencing reads were aligned to the UCSC ce10 *C. elegans* reference genome using HISAT2 (*Kim et al., 2015*). Reads aligning to genetic features were then counted using HTSeq-count (*Anders et al., 2015*) and analyzed for differential expression analysis using DESeq2 (*Love et al., 2014*). For analysis shown in *Figure 3C*, differential expression analysis was done using Tophat (V.2.0.8) and Cufflink (V.2.0.2). Genes differentially expressed in wild type vs *meg-3 meg-4* embryos are listed in *Supplementary file 3*. The command lines for Tuxedo suit are listed as below:

For each biological sample, sequencing reads were first mapped to ce10 reference genome using tophat2:

```
$ tophat2 –p 12 g 1 –output-dir segment-length 20 –min-intron-length 10 –
max-intron-length 25000 G < gene.gtf> –transcriptome-index<Name.fastq>
```

For differential gene expression analysis, sets of independent mutant and control mapped reads (e.g biological replicates) were used in cuffdiff analysis:

```
$ cuffdiff -p 12 -o < output >  compatible-hits-norm –upper-quartile-norm -
b < genome.fa><genes.gtf><tophat output_sample 1, tophat output_sample 2, tophat
output_sample 3,..><tophat output_control1, tophat output_control2, tophat out-
put_control3,.. >
```

## Ribosome profiling

Libraries for wild type and *meg-3 meg-4* embryos were trimmed to remove the ligated 3' linker (C TGTAGGCACCATCAAT) with skewer (*Jiang et al., 2014*). For the rest of our libraries, the 3' adapter (NNNNNNCACTCGGGCACCAAGGA) was trimmed, and four random nucleotides included in the RT primer (RNNNAGATCGGAAGAGCGTCGTGTAGGGAAAGAGTGTAGATCTCGGTGGTCGC/iSP18/TTCAGACGTGTGCTCTTCCGATCTGTCCTTGGTGCCCGAGTG) were removed from the 5' end of reads. Trimmed reads longer than 15 nt were aligned to reference genome ce10 using STAR (*Dobin et al., 2013*) with '–outFilterMismatchNoverLmax 0.3'. Unmapped reads were then mapped to genome using the following options '–outFilterIntronMotifs RemoveNoncanonicalUnannotated –outFilterMultimapNmax 1 –outFilterMismatchNoverLmax 0.1'. Aligned reads were than counted and analyzed using HTseq-count (*Anders et al., 2015*), DEseq2 (*Love et al., 2014*) and custom R code (RStudio 1.2). Differential translation efficiency between wild type and *meg-3 meg-4* was analyzed by the Riborex R package (*Li et al., 2017*), the results of which are listed in *Supplementary file 4*. Ribosome footprints for P-blastomere enriched genes are listed in *Supplementary file 2*. Correlation analyses of sequencing libraries are shown in *Figure 2—figure supplement 3*.

## Gene list: MEG-3 bound and P granule transcripts

Read counts obtained in the control GFP iCLIP were used to define background threshold (read count = 60.) We defined MEG-3 bound transcripts by excluding transcripts with MEG-3::GFP iCLIP read counts < 60 as shown in *Figure 1B* (stippled horizontal line at $\log_e 60 = 4.1$). To define P granule transcripts, we used the rank order of *F35G2.1* (Rank 388), the left most gene in the *nos-2* cluster as shown in *Figure 1D*, as the cut off. Genes with rank order better than 388 in either one of MEG-3::GFP iCLIP were defined as P granule transcripts (*Supplementary file 1*).

## Single molecule fluorescence in situ hybridization (smFISH)

smFISH probes were designed using Biosearch Technologies's Stellaris Probe Designer. The fluorophores used in this study were Quasar570 and Quasar670. For sample preparation, embryos were extruded from adults on poly-lysine slides (0.01%) and subjected to freeze-crack followed by cold methanol fixation at −20° C. Samples were washed five times in PBS+0.1%Tween20 and fixed in 4% PFA (Electron Microscopy Science, No.15714) in PBS for one hour at room temperature. Samples were again washed four times in PBS+0.1%Tween20, twice in 2x SCC, and once in wash buffer (10% formamide, 2x SCC) before blocking in hybridization buffer (10% formamide, 2x SCC, 200 ug/mL BSA, 2 mM Ribonucleoside Vanadyl Complex, 0.2 mg/mL yeast total RNA, 10% dextran sulfate) for 30 min at 37° C. Hybridization was then conducted by incubating samples with 50–100 nM probe solutions diluted in hybridization buffer overnight at 37° C. Following hybridization, samples were washed twice in wash buffer at 37° C, twice in 2x SCC, once in PBS-Tween20 (0.1%), and twice in PBS. Lastly, samples were mounted using VECTASHIELD Antifade Mounting Media with DAPI or Prolong Diamond Antifade Mountant.

## Confocal microscopy

Fluorescence confocal microscopy was performed using a Zeiss Axio Imager with a Yokogawa spinning-disc confocal scanner. Embryo images were taken using Slidebook v6.0 software (Intelligent

Imaging Innovations) using a 63x objective. Embryos were staged by DAPI-stained nuclei in optical Z-sections and multiple Z-sections were taken to include germ cells. For In vitro condensation reactions, images are single planes taken using a 40x objective unless otherwise indicated. For fluorescence super-resolution microscopy, images were acquired using ZEISS LSM 880-AiryScan (Carl Zeiss) equipped with a 63X objective. Images were processed using ZEN imaging software (Carl Zeiss). Equally normalized images were exported via either Slidebook v6.0 or ZEN, and contrasts of images were equally adjusted between control and experimental sets. For in vitro fluorescence recovery after photobleaching experiments, images were acquired using Zeiss LSM 800 GaAsp. Images are single confocal planes imaged using a 63x objective every 3 s during a recovery phase of 300 s. All image analyses were conducted using the Fiji image-processing package (http://fiji.sc/Fiji).

## Quantification of RNA granule size

For measurements reported in *Figure 1D*, 9 Z-planes (0.8 µm step size) from the center of an embryo were extracted for image analysis. To identify RNA granules, minimum and maximum values for thresholding were set as follows: minimum = the mean intensity of the background signal plus one standard deviation of the background intensity; maximum = the mean intensity of the background signal plus six standard deviations of the background intensity. After thresholding, the nucleus counter cookbook plugin in FIJI was used to identify RNA granules in germ cells. Objects of less than two pixels were filtered out to minimize noise, a watershed filter was applied to improve separation of granule signals close in proximity, and the image was converted to a binary image by the 'Current' method. Measurements for granule area were extracted from the ROI manger. For *Figure 1D*, mean and standard deviation of granule size from at least four embryos were plotted against average read counts from two MEG-3::GFP iCLIP experiments.

## Temperature shifts

Temperature shift experiments were performed by transferring gravid worms grown at 20°C to pre-warmed 30°C plates for 15 min. After heat shock, worms were immediately dissected for smFISH experiments or live imaging. Under these conditions, the stress granule marker G3BP coalesces into granules in all cells, some of which associate with P granules (*Figure 2—figure supplement 2E*). To quantify the ratio of smFISH signal in MEG-3::GFP granules, eight single Z planes were extracted and used for image analysis as shown in *Figure 2E*.

MEG-3::GFP granules were identified using the nucleus counter plugin as described above. The minimum threshold was set to the two times the mean intensity of the background signal of the image; and the maximum threshold was calculated by adding six standard deviations of the background intensity. A mask generated from objects identified by the nucleus counter plugin was applied to the raw image to extract RNA or GFP intensity. To remove background signal, the mean intensity of an object across the nucleus was measured, and subtracted from calculated RNA or GFP intensities. To calculate the ratio of RNA signal/MEG-3::GFP granule signal in each selected Z plane, the sum of intensities of RNA in identified objects ($I_{smFISHg}$) was divided by the total intensity of the MEG-3::GFP in the same objects ($I_{meg3g}$). The intensity ratio of RNA in MEG-3 granules ($I_{smFISHg}/Imeg_{meg3g}$) before and after heat shock was compared. Each data point represents data from one Z plane acquired from two embryos (eight planes per embryo).

## Translation inhibitor treatments

For drug treatment, *ptr-2* RNAi was used to permeabilize the egg shell. A 12.5 mg/mL (20X) Puromycin stock solution (sigma, P8833) was made with osmolarity calibrated Egg buffer [118 mM NaCl, 48 mM KCl, 2 mM CaCl$_2$, 2 mM MgCl$_2$ 25 mM HEPES pH 7.3, 340 ± 5 mOsm]. A 50 mg/mL (200X) Cycloheximide stock solution was made in ethanol. RNAi-treated gravid worms were dissected and permeabilized embryos were released into drug containing egg buffer for 1 hr in a humidity chamber to maintain vapor pressure. Both puromycin and cycloheximide induced cell cycle arrest. After drug treatment, excess buffer was removed and embryos were subjected to image acquisition and quantification using nucleus counter cookbook plugin as described in 'Quantification of RNA granule size' section above. Images used for quantification are maximum Z projections acquired using a 63x oil.

## RNA enrichment in germ granule vs cytosol

smFISH quantification was conducted using Imaris Image Analysis Software visualization in 3D space. The boundary/volume for the germ cell cytosol and germ granules was created by Surface function using MEG-3::GFP signal in Imaris. The sum of intensity of the germ cell cytosol and granules were extracted and the percentage of RNA enrichment in germ granules was calculated.

## Immunostaining

As in sample preparation for smFISH experiments, embryos were extruded from adults and subjected to freeze-crack on poly-lysine slides followed by cold methanol fixation for 15 min and then cold acetone for 10 min. Slides were blocked twice in PBS-Tween20 (0.1%)-BSA (0.1%) for 30 min at room temperature, and incubated with 90 µl primary antibody overnight at 4°C in a humidity chamber. Antibody dilutions (in PBST/BSA): mouse K76 1:10 (DSHB), Rat α-OLLAS 1:80 (Gift from Dr. Jeremy Nathans), mouse α-FLAG M2 1:500 (Sigma F3165). Secondary antibodies (Molecular Probes/ Thermo Fisher Sci.) were applied for 1 ~ 2 hr at room temperature.

## RNA extraction and preparation of mRNA-seq library preparation

RNA was extracted from embryos or cleared embryo lysates using TRIZOL. The aqueous phase was transferred to Zymo-SpinTM IC Column (Zymo research R1013) for concentration and DNase I treatment as described in manual. RNA quality was assayed by Agilent Bioanalyzer using Agilent RNA 6000 Pico Chip. All RNAs used for library preparation had RIN (RNA integrity number)>9. For mRNA-seq library construction, 0.5 µg of total RNA was treated with Ribo-Zero Gold Epidemiology rRNA Removal Kit. Libraries were then prepared following the TruSeq RNA Library Prep Kit v2 instruction. All sequencing was performed using the Illumina HiSeq2500 at the Johns Hopkins University School of Medicine Genetic Resources Core Facility.

## Protein purification and labeling

### Purification of MEG-3 His-tagged fusion

MEG-3 full-length (aa1-862) fused to an N-terminal 6XHis tag in pET28a was expressed and purified from inclusion bodies using a protocol modified from *Smith et al. (2016)*; *Putnam et al. (2019)* to improve purity and yield. MEG-3 was grown in Rosetta (DE3) cells at 37°C in terrific broth + ampicillin (100 µg/mL) to an OD600 of ~1.0 and induced with 1 mM IPTG at 16° C for 16 hr. Cells were resuspended in Buffer A (20 mM HEPES pH 8.0, 1000 mM KCl, 10% (vol/vol) glycerol, 0.5% Triton-X100, 2 mM DTT, 0.4 mM PMSF, and Roche proteinase inhibitors), lysed by sonication, and spun at 13,000 rpm for 30 min. The supernatant was discarded and the pellet containing MEG-3 inclusion bodies was resuspended in Buffer A, briefly sonificated, and spun at 13,000 rpm. The pellet was solubilized overnight at 4°C in Buffer A with 6 M Urea. The solubilized protein was filtered (0.45 µm), and passed over a HisTRAP 5 mL column (GE Healthcare). Bound protein was washed with Buffer B (20 mM HEPES pH 8, 1 M KCl, 25 mM Imidazole pH 6, 10% (vol/vol) glycerol, 6 M urea, 2 mM DTT) and eluted in Buffer C (20 mM HEPES pH 7.4, 1 M KCl, 250 mM Imidazole pH 6, 10% (vol/vol) glycerol, 6 M urea, 2 mM DTT). Protein containing fractions were concentrated to 3 mL and further purified by size exclusion using a HiPrep 16/60 Sephacryl S-500 HR (GE Healthcare) in Buffer D (20 mM HEPES pH 7.4, 1 M KCl, 10% (vol/vol) glycerol, 6 M urea, 2 mM DTT). Aliquots of peak elution fractions were run on 4–12% Bis Tris gels, and stained with Simply Blue Safe Stain (ThermoFisher Waltham, MA). Protein was concentrated to a final concentration of 2–5 mg/mL, aliquoted, snap frozen in liquid nitrogen, and stored at −80°C (*Figure 4—figure supplement 1A*)

### Purification of PGL-3

MBP-TEV-PGL-3 was expressed and purified as described (*Putnam et al., 2019*) with the following modifications: MBP was cleaved using homemade TEV protease instead of commercial. A plasmid expressing 8X-His-TEV-8X-Arg tag protease was obtained from Addgene and purified according to the published protocol (*Tropea et al., 2009*). Before loading cleaved PGL-3 protein on to a heparin affinity matrix, cleaved MBP-6X-His and 8X-His-TEV protease were removed using a HisTRAP column (GE Healthcare).

Protein labeling

Proteins were labeled with succinimidyl ester reactive fluorophores from Molecular Probes (Alexa Fluor 647, Alexa Fluor 555, or DyLight 488 NHS Ester) following manufacturer's instructions. Free fluorophore was eliminated by passage through three Zeba Spin Desalting Columns (7K MWCO, 0.5 mL) into protein storage buffer. The concentration of fluorophore-labeled protein was determined using fluorophore extinction coefficients measured on a Nanodrop ND-1000 spectrophotometer. Labeling reactions resulted in ~0.25–1 label per protein. Aliquots were snap frozen and stored. In condensation experiments, fluorophore-labeled protein was mixed with unlabeled protein for final reaction concentrations of 25–100 nM of fluorophore labeled protein.

## In vitro RNA preparation

mRNAs were transcribed using T7 or SP6 mMessageMachine (Thermofisher) using manufacturer's recommendation. 1 μL of ChromaTide Alexa Fluor 488–5-UTP or 546–14-UTP (Thermofisher) were added to transcription reactions to fluorescently trace label mRNAs. Template DNA for transcription reactions was obtained by PCR amplification from plasmids. *nos-2* fragments were generated by PCR amplification from the 5' end of the full length *nos-2* template DNA. Free NTPs and protein were removed by lithium chloride precipitation. RNAs were resuspended in water and stored at −20°C. The integrity of RNA products was verified by agarose gel electrophoresis.

30 nt oligo polyU RNAs were ordered from IDT either unlabeled or with a 3' FAM modification. Oligos were resuspended in water aliquoted and stored at −80°C. Labeled and unlabeled oligo RNAs were mixed together and used at final concentrations of 20 ng/uL including 25 nM fluorescently labeled oligo.

## In vitro condensation experiments and analysis

Protein condensation was induced by diluting proteins out of storage buffer into condensation buffer containing 25 mM HEPES (pH 7.4), salt adjusted to a final concentration of 150 mM (37.5 mM KCl, 112.5 mM NaCl), and RNA. Unless otherwise indicated, for all co-assembly experiments, we used 500 nM MEG-3 and 20 ng/μL RNA. MEG-3 solutions contained 25 nM fluorescent trace labels with either 488, 555, or 647 (indicated in figure legends). Condensate reactions with the RNA dye contained a final concentration of 100 nM SYTO 14. Condensation reactions were incubated at room temperature for 30 min or as indicated, before spotting onto thin chambered glass slides (ERIE SCIENTIFIC COMPANY 30-2066A) with a coverslip. Images used for quantification are single planes acquired using a 40x oil objective over an area spanning 171 × 171 μm.

To quantify the ratio of protein or RNA in condensates, a mask was created by thresholding images, filtering out objects of less than four pixels to minimize noise, applying a watershed filter to improve separation of objects close in proximity, and converting to a binary image by the Otsu method using the nucleus counter cookbook plugin. Minimum thresholds were set to the mean intensity of the background signal of the image plus 1–2 standard deviations. The maximum threshold was calculated by adding 3–4 times the standard deviation of the background. Using generated masks, the integrated intensity within each object was calculated. To remove non-specific background signal the mean intensity of an image field in the absence of the labeled component was subtracted from each pixel yielding the total intensity of each object.

Histograms of MEG-3 intensity were generated by taking the log(10) of total intensity for each MEG-3 object, Log(I). Objects in 2–3 experimental replicates of 4 images were identified and quantified as described above the number of objects for each Log(I) value was binned (bin size = 0.2 Log(I) units), and normalized to the total number of objects. The percent of objects in each bin was averaged for each experimental replicate (*Figure 4B*, *Figure 4—figure supplement 2E*, *Figure 5—figure supplement 1A*).

The percent of MEG-3 in aggregates or condensates was determined by comparing histograms of reactions in which all objects are aggregates (500 mM NaCl or no RNA) or condensates (150 mM NaCl and 20–80 ng/μL *nos-2* RNA) as illustrated in *Figure 4B*, *Figure 4—figure supplement 2E*, *Figure 5—figure supplement 1A*. The minimum at the intersection of the two conditions was calculated. The percentage of MEG-3 objects with an intensity above or equal to the intersection were classified as aggregates and the fraction of objects with an intensity below the intersection were classified as condensates.

To calculate the fraction of RNA in MEG-3 condensates/aggregates, the background corrected sum of RNA fluorescent intensity in each MEG-3 object was divided by the total intensity of RNA fluorescence in the imaged area (*Figure 4C*).

Radii of MEG-3 condensates were estimated by imaging condensation reactions of 500 nM MEG-3 and 40 ng/µL RNA (*Figure 4—figure supplement 2A*). Images used for quantification were single planes acquired using a 100x oil objective over an area spanning 68 × 68 µm. four experimental replicates of 16 images were identified and quantified (>1500 objects/replicate) as described above, and radii were calculated from the area of each object. Calculated radii are an overestimate and represent upper limits for actual condensate size. The number of objects for each radii was binned (bin size = 0.06 µm), and normalized to the total number of objects. The percent of objects in each bin was averaged for each experimental replicate (*Figure 4—figure supplement 2B*).

## Fluorescence Recovery after Photobleaching (FRAP)

20 µL condensation reactions (prepared as described above) were added to a chambered coverglass (Grace BioLabs) and imaged using a Zeiss LSM 800 GaAsp. Bleaching was performed using 100% laser power in the 488, 546, or 647 channels. Regions slightly larger than the condensates (radius ≈ 3 µM) were photobleached. A single confocal plane was imaged using a 63x objective every 3 s during a recovery phase of 300 s.

FRAP analysis was performed as described in *Putnam et al. (2019)*. Briefly, fluorescence recovery was corrected for background and normalized to the initial granule intensity using the equation: $nI = (I-I^{bkg})/(I^i-I^{bkgi})$, where $nI$ is the background corrected and normalized fluorescence intensity, $I$ is the intensity of the FRAPed granule, $I^{bkg}$ is the fluorescence intensity outside of the condensate, $I^i$ is the initial intensity before bleaching, and $I^{bkgi}$ is the initial background intensity. Recovery rates were determined by fitting individual traces to a first order equation $nI = (A^{rec}·(1-e^{-kt})$, where $A^{rec}$ is the fluorescence recovery amplitude and $k$ is the rate of fluorescence recovery. For RNAs where fluorescence recovery was in the linear range for the entire time course, initial recovery rates were calculated by fitting to a linear equation $nI = kt$, where $k$ is the initial rate of fluorescence recovery (*Figure 5D*, *Figure 5—figure supplement 1B*).

## Ex vivo extrusion experiments

1 mM SYTO 14 RNA Dye (ThermoFisher) and 1 mM JF$_{646}$ (JF$_{646}$, Janelia *Grimm et al., 2015*) dissolved in DMSO were diluted 500 fold into osmolarity calibrated Egg buffer [118 mM NaCl, 48 mM KCl, 2 mM CaCl$_2$, 2 mM MgCl$_2$25 mM HEPES pH 7.3, 340 ± 5 mOsm] to reach 2 µM stock solutions. *perm-1(*RNAi) adult gravid hermaphrodites expressing MEG-3::HALO (*Wu et al., 2019*) were dissected and egg shell permeabilized embryos were released into 10 µL egg buffer containing 1 µM SYTO 14 and JF$_{646}$ for 10 min in a humid chamber to prevent evaporation. After drug treatment, embryos were washed 3X with egg buffer without drug. Approximately 200–20 µm polysterene beads (Bangs Laboratories) suspended in egg buffer were added to prevent embryo compression, and placed on slides for imaging. Embryo contents were extruded by puncturing the eggshell near the anterior region of the germline blastomere using a 3i Ablate! laser system at 532 nm pulse setting with a power level of 155 (*Putnam et al., 2019*). All embryo images are Z stack maximum projections using a Z step size of 1 µm, spanning the depth of the embryo. Images were acquired in the 488 and 647 channel every 10 s using a 63x objective (*Figure 5E*, *Figure 5—figure supplement 1C*).

To quantify SYTO 14 and MEG-3::Halo persistence in granules, photomicrographs acquired as described above were analyzed using FIJI. Not all SYTO 14 granules were MEG-3::Halo positive, and are potentially P bodies (*Gallo et al., 2008*). Only SYTO 14 granules also positive for MEG-3::Halo were quantified. MEG-3::Halo granules were identified using the nucleus counter plugin as described above. The total intensity of objects was quantified for both 488 (SYTO 14) and 647 (Halo) channels. Total fluorescence intensity was calculated before ($I_B$) and after ($I_A$) extrusion and used to calculate a fluorescence ratio ($I_A/I_B$). Photobleaching was minimal for MEG-3::Halo; however it was significant for SYTO 14. To correct for photobleaching, total fluorescence intensity was corrected for photobleaching using the average photobleaching rate calculated from the cytoplasm of intact embryos in the imaging area. For some embryos, granules left the field of view and could not be counted. The $I_A/I_B$, therefore, is a minimal estimate of the fraction MEG-3 and SYTO 14 that remained in the granule phase after extrusion (*Figure 5F*).

## Graphing and data fitting

All data were plotted and statistical analysis was conducted using Graphpad Prism seven software. Fitting of recovery curves in FRAP experiments was conducted using Kaleidagraph (Synergy) software.

## Data and materials availability

Sequencing datasets and processed results generated in this paper are available at GEO accession GSE139881 for iCLIP (GSE139878), embryonic RNAseq (GSE139879) and ribosome profiling results (GSE139880).

# Acknowledgements

We thank the Johns Hopkins Neuroscience Research Multiphoton Imaging Core (NS050274) and the Johns Hopkins Integrated Imaging Center (S10OD023548) for excellent microscopy support. We thank the Lavis lab for Halo Ligand JF$_{646}$, the Griffin lab for MEG-3::Halo strain, the Nathans lab for the OLLAS antibody, Addgene for TEV protease from the Waugh lab, and Colin Wu for assistance with ribosome profiling experiments. We especially thank Dr. Sijung Yun (Yotta Biomed) for his assistance with data analysis. We also thank the Baltimore Worm club and the Seydoux lab for many helpful discussions. This work was supported by the National Institutes of Health (NIH) (grant number R37 HD37047). JPTO was supported by the JHU SOM Biochemistry, Cellular, and Molecular Biology NIH training grant (T32 GM007445). GS is an investigator of the Howard Hughes Medical Institute. Some strains were provided by the CGC, which is funded by NIH Office of Research Infrastructure Programs (P40 OD010440). Data acquired using the Zeiss LSM 800 Confocal reported in this publication was supported by Office of the Director, NIH (OD) of the National Institutes of Health (award number S10OD016374).

# Additional information

## Competing interests

Geraldine Seydoux: serves on the Scientific Advisory Board of Dewpoint Therapeutics, Inc. The other authors declare that no competing interests exist.

## Funding

| Funder | Grant reference number | Author |
| --- | --- | --- |
| National Institutes of Health | R37 HD37047 | Geraldine Seydoux |
| Howard Hughes Medical Institute | | Geraldine Seydoux |
| National Institutes of Health | T32 GM007445 | John Paul T Ouyang |

The funders had no role in study design, data collection and interpretation, or the decision to submit the work for publication.

## Author contributions

Chih-Yung S Lee, Conceptualization, Data curation, Software, Formal analysis, Supervision, Validation, Investigation, Visualization, Methodology; Andrea Putnam, Conceptualization, Data curation, Software, Formal analysis, Validation, Investigation, Visualization, Methodology; Tu Lu, Data curation, Software, Formal analysis, Validation, Visualization; ShuaiXin He, John Paul T Ouyang, Data curation, Visualization; Geraldine Seydoux, Conceptualization, Supervision, Funding acquisition, Investigation, Project administration

## Author ORCIDs

Andrea Putnam (iD) https://orcid.org/0000-0001-7985-142X
Tu Lu (iD) http://orcid.org/0000-0002-5697-300X

John Paul T Ouyang (iD) https://orcid.org/0000-0003-2030-432X
Geraldine Seydoux (iD) https://orcid.org/0000-0001-8257-0493

## Decision letter and Author response

Decision letter https://doi.org/10.7554/eLife.52896.sa1
Author response https://doi.org/10.7554/eLife.52896.sa2

# Additional files

## Supplementary files

• Supplementary file 1. Gene list of MEG-3 bound transcripts, P granule transcripts and PGL-1 bound transcripts.

• Supplementary file 2. Ribosome footprint for P-blastomere enriched genes.

• Supplementary file 3. Differential gene expression analysis of wild type and *meg-3meg-4* embryos.

• Supplementary file 4. Differential Translation analysis of wild type and *meg-3meg-4* embryos.

• Supplementary file 5. Sequencing library information.

• Supplementary file 6. Read counts for iCLIP experiments.

• Transparent reporting form

## Data availability

Sequencing data have been deposited in GEO under accession number GSE139881,GSE139878, GSE139879 and GSE139880. Description of iCLIP analysis and additional python codes are deposited in Github: https://github.com/fishhead1978/iCLIP_2019 (copy archived at https://github.com/elifesciences-publications/iCLIP_2019).

The following datasets were generated:

| Author(s) | Year | Dataset title | Dataset URL | Database and Identifier |
|---|---|---|---|---|
| Lee C-Y, Seydoux G | 2019 | Recruitment of mRNAs to P granules by gelation with intrinsically-disordered proteins (iCLIP results | https://www.ncbi.nlm.nih.gov/geo/query/acc.cgi?acc=GSE139878 | NCBI Gene Expression Omnibus, GSE139878 |
| Lee C-Y, Seydoux G | 2019 | Recruitment of mRNAs to P granules by gelation with intrinsically-disordered proteins (RNAseq and ribosome profiling) | https://www.ncbi.nlm.nih.gov/geo/query/acc.cgi?acc=GSE139880 | NCBI Gene Expression Omnibus, GSE139880 |
| Lee C-Y, Seydoux G | 2019 | Recruitment of mRNAs to P granules by gelation with intrinsically-disordered proteins (RNAseq datasets) | https://www.ncbi.nlm.nih.gov/geo/query/acc.cgi?acc=GSE139879 | NCBI Gene Expression Omnibus, GSE139879 |

The following previously published dataset was used:

| Author(s) | Year | Dataset title | Dataset URL | Database and Identifier |
|---|---|---|---|---|
| Lee C-Y, Seydoux G | 2017 | Chromatin reprogramming in primordial germ cells requires Nanos-dependent down-regulation of the synMuvB transcription factor LIN-15B | https://www.ncbi.nlm.nih.gov/geo/query/acc.cgi?acc=GSE100652 | NCBI Gene Expression Omnibus, GSE100652 |

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
