## [Decision Letter]

**Acceptance summary:**

RNA granules are protein-RNA condensates that are not enclosed by membranes. In this paper, Seydoux and colleagues use a variety of techniques to examine the messenger RNA content of P granules. P granules are present in germline cells in *C. elegans*. Significant findings include the characterization of mRNAs that are bound in P granules and the demonstration that P granule formation is driven by the intrinsically disordered protein MEG-3. Similarities between P granules and stress granules are also revealed by these studies.

**Decision letter after peer review:**

Thank you for submitting your article "Recruitment of mRNAs to P granules by gelation with intrinsically-disordered proteins" for consideration by *eLife*. Your article has been reviewed by James Manley as the Senior Editor, a Reviewing Editor, and two reviewers. The following individuals involved in review of your submission have agreed to reveal their identity: Roy Parker (Reviewer #1); Dustin L Updike (Reviewer #2).

The reviewers have discussed the reviews with one another and the Reviewing Editor has drafted this decision to help you prepare a revised submission.

The reviewers concurred that your manuscript presented significant results and was in principle suitable for *eLife*. Nevertheless, both referees made a number of comments regarding presentation. They also pointed out a number of specific issues that deserve clarification. Finally, they suggest a couple of straightforward experiments that would strengthen the paper. Please address these issues as thoroughly as possible.

*Reviewer #1:*

This manuscript addresses the mechanisms by which RNAs and proteins condense together to form P-granules, and what the function of P-granules is in terms of regulating mRNA function. The work makes two main points. First, it demonstrates that P-granules are formed from untranslating mRNPs that interact in a sequence independent manner with the Meg proteins. This is a nice demonstration of the composition and assembly mechanisms of P-granules and synergizes well with what is known about the selectivity of RNA recruitment to other RNP granules. Second, the work demonstrates that the function of P-granules in development is to promote the localization of specific mRNAs into the P4 cell. This is a very significant contribution. Taken together, this work is an important manuscript and after addressing the comments below should be published in *eLife*.

Essential revisions:

1) I think the most important contribution of the manuscript is the demonstration of a clear function for P-granules in the accumulation of mRNAs in the P4 cell. Given this, I suggest:a) Change the title to something like: "P-granules function to drive cell-type specific accumulation of mRNAs".b) Rewrite the abstract to emphasize this point (or at least mention it).c) Add at least one paragraph in the discussion that makes this point, and supports it with the critical observations. Ideally, this would be followed by second paragraph discussing how the formation of a P-granule creates a localization sink by changing the diffusion rate of granules as compared to individual mRNAs.

2) Although the authors present convincing data that Meg3/4 are required for P-granule assembly in embryos and can form assemblies in vitro with Meg-3 protein, I would recommend the authors be cautious about their interpretation that "P-granules from by gelation with intrinsically disordered proteins" for a few reasons.a) How do the authors know the assembly is not driven by RNA interactions, and then that RNA assembly recruits Meg-3 to it? At a minimum, the authors should examine the assembly/"gelation" of RNA in their in vitro conditions in the absence of the Meg-3 protein to assure themselves that the RNA-Meg-3 assemblies they study require Meg-3 for assembly.b) Since Meg-3 forms "aggregates" by itself, which are solubilized by RNA, it seems that RNA is altering Meg-3 self-interactions. The manuscript would be improved if the authors could clarify if these Meg-3 aggregates are artifacts of the in vitro system, or do they have biological meaning?c) What is the difference between gelation and assembly in this context? As I see it, the authors show that the RNA and protein assemblies formed in vitro show slow rates of Meg-3 or RNA exchange in a manner affected by RNA length. Why not just call these non-dynamic assemblies since "gels" implies different things to different people. If they are defining an assembly as a gel if it has a slow exchange rate of a component, then they should be explicit about this point. If they are defining a gel as an assembly where every component is interconnected, then referring to these assemblies as gels seems premature.

3) The work in this manuscript is consistent with Meg3/4 playing a role in promoting RNP condensation when there is a limited amount of untranslating mRNAs. A prediction of this model is that when cells are heat shocked, the increase in untranslating mRNAs will correspondingly drive RNP granule formation. Thus, one predicts that in a meg3/4 mutant, heat shock would restore P-granules. (Similar results have been seen with P-bodies in mammalian cells, where lsm4 RNAi abolishes P-bodies, but when cells are then stressed, P-bodies rapidly reassemble (Kedersha et al., 2005). Given that this is a simple experiment, and if it worked would reveal a new property of P-granules, it would be nice to try, although it is not required for publication.

3b) A related issue is that it would improve the manuscript to clarify if separate stress granules form under heat shock, or those that form, merge with P-granules.

4) As I read it, the analysis of mRNAs in Figure 1D that are called as "enriched" in P-granules will also be strongly affected by their abundance. As such, this analysis could miss low abundance but highly enriched mRNAs. I suggest either a) clarify how the analysis was done such that this is not an issue, or b) acknowledge the limitations in the text with how this analysis is performed and the types of RNAs it will miss.

*Reviewer #2:*

Seydoux and Fire, (1994) showed the enrichment of polyadenylated transcripts in the germline blastomeres of the early embryo; after, a handful of developmentally regulated mRNAs, but not rRNAs, were shown to be enriched in the P granules of early embryos (Subramaniam, 1999; Schisa, 2001). The identity of P-granule bound mRNAs is instrumental to understanding P-granule function in the early embryo, but in the intervening years, this small list of P-granule transcripts has not increased despite attempts from multiple labs.

In this report, Chih-Yung Lee and colleagues in the Seydoux lab break through this barrier by identifying P-granule transcripts through MEG-3 iCLIP. MEG-3 forms small non-dynamic condensates within P granules, in contrast to PGL protein condensates that are highly dynamic. The authors identify just 18 PGL-1-bound transcripts compared to 657 that bound to MEG-3. 18/18 transcripts from this list localize robustly to P granules, and this localization is MEG-3 dependent.

The authors show that MEG-3 transcripts demonstrate no apparent sequence conservation, but correlate with a low-occupancy of ribosomes, suggesting that untranslated transcripts have an affinity for P granules. They then show that translational inhibition recruits new transcripts to P granules. Upon translational activation in the germline precursor cell (P4) of nos-2 and Y51F10.2, these transcripts disperse from P granules to localize throughout the cytoplasm. The timing of nos-2 and Y51F10.2 translation is unaffected in meg-3 meg-4 mutants, suggesting that P-granule localization is a consequence of translational repression and that P granules have some role in delivering transcripts that encode germ cell fate regulators to the nascent germline. Recombinant MEG-3 was shown to interact with RNA through electrostatic interactions in a non-sequence specific, but length-dependent manner and these stable interactions persist in both in vitro and ex vivo conditions.

This paper presents several new and significant advances to the RNP phase-separation and germ-granule fields. Two crucial but addressable criticisms are discussed.

1) As a control for the MEG-3 iCLIP experiments, the authors perform iCLIP on GFP-expressing embryos. Information for the control (JH1094?) isn't given. The optimal control would express GFP in just the germline blastomeres where MEG-3 and PGL-1 are enriched to avoid identifying the most abundant transcripts in P blastomeres (compared to the whole embryo). Genes like cbd-1, cgh-1, and puf-5 are some of most abundant in the germline. Of the 657 MEG-3-bound transcripts, over half are in the top 1000 germline expressed (11-fold more than expected by chance), and most fall within the top 3000 germline-expressed genes. The statement on line 102 that MEG-3 binds to a specific subset of mRNAs instead of abundant P-blastomere mRNAs will need more supporting data than is currently provided.

Correlations provided in Figure 1C and Figure 1—figure supplement 1C may be erroneous and serve as a distraction at the beginning of an otherwise very significant study. Instead, the comparison should be with all ~ 15,000 germline-expressed transcripts. Despite a potential lack of MEG-3 specificity, results are still in line with the author's later and well-supported observations that untranslated transcripts in translationally quiescent P blastomeres have an affinity for MEG-3.

An apparent exception to the transcript abundance correlation is that MEG-3-bound transcripts lack those that encode ribosomal proteins. While not necessary to include in this manuscript, it may be interesting to consider why abundant RPL/RPS (RPG) transcripts do not bind MEG-3.

2) P blastomeres are translationally quiescent, so shouldn't lower ribosomal coverage of MEG-3 attached transcripts in P blastomeres be expected when the experiment was done on whole embryos? Specificity is not addressed and observations in Figure 2C may be a correlation artifact. While this isn't likely given the later in situ hybridization experiments following translational inhibition, the caveat should be discussed.

---

## [Author Response]

Reviewer #1:Essential revisions:1) I think the most important contribution of the manuscript is the demonstration of a clear function for P-granules in the accumulation of mRNAs in the P4 cell. Given this, I suggest:a) Change the title to something like: "P-granules function to drive cell-type specific accumulation of mRNAs".b) Rewrite the abstract to emphasize this point (or at least mention it).c) Add at least one paragraph in the discussion that makes this point, and supports it with the critical observations. Ideally, this would be followed by second paragraph discussing how the formation of a P-granule creates a localization sink by changing the diffusion rate of granules as compared to individual mRNAs.

We have modified the abstract to highlight our findings with regard to P granule function and have reworked the Discussion section “P granules enrich maternal mRNAs in the germline founder cell P_4_ to maximize the robustness of germ cell fate specification”. Another important point of the paper is the demonstration of a direct role for an intrinsically-disordered protein in recruiting RNAs to RNA granules. To our knowledge this has not yet been shown for other RNA granules, and so we are choosing to maintain the original title.

2) Although the authors present convincing data that Meg3/4 are required for P-granule assembly in embryos and can form assemblies in vitro with Meg-3 protein, I would recommend the authors be cautious about their interpretation that "P-granules from by gelation with intrinsically disordered proteins" for a few reasons.a) How do the authors know the assembly is not driven by RNA interactions, and then that RNA assembly recruits Meg-3 to it? At a minimum, the authors should examine the assembly/"gelation" of RNA in their in vitro conditions in the absence of the Meg-3 protein to assure themselves that the RNA-Meg-3 assemblies they study require Meg-3 for assembly.

We include a new figure (Figure 4—figure supplement 1B) which shows that RNA alone does not form condensates under our assay conditions. Similarly, in vivo, *nos-2* and *Y51* RNAs do not accumulate in macroscopic granules in embryos lacking *meg-3meg-4*. We are confident therefore that MEG proteins promote RNA condensation. We are not excluding the possibility that smaller RNA aggregates form independently of *meg-3meg-4* and refer to the possibility of RNA-only aggregation in the Introduction.

b) Since Meg-3 forms "aggregates" by itself, which are solubilized by RNA, it seems that RNA is altering Meg-3 self-interactions. The manuscript would be improved if the authors could clarify if these Meg-3 aggregates are artifacts of the in vitro system, or do they have biological meaning?

We estimate that, in embryos, the stochiometric ratio of MEG-3 to mRNA molecules is ~1:5. MEG-3 aggregates form in vitro under much higher MEG-to-RNA stochiometric ratios (e.g. 1:0.05). MEG-3 aggregates, therefore, may not be physiological. However, because we do not know how much RNA is available to MEG-3 in vivo, we prefer not to make any conclusions re: the physiological relevance of MEG-3 aggregates at this time.

c) What is the difference between gelation and assembly in this context? As I see it, the authors show that the RNA and protein assemblies formed in vitro show slow rates of Meg-3 or RNA exchange in a manner affected by RNA length. Why not just call these non-dynamic assemblies since "gels" implies different things to different people. If they are defining an assembly as a gel if it has a slow exchange rate of a component, then they should be explicit about this point. If they are defining a gel as an assembly where every component is interconnected, then referring to these assemblies as gels seems premature.

We have replaced “gel” with “non-dynamic assemblies” throughout the manuscript.

3) The work in this manuscript is consistent with Meg3/4 playing a role in promoting RNP condensation when there is a limited amount of untranslating mRNAs. A prediction of this model is that when cells are heat shocked, the increase in untranslating mRNAs will correspondingly drive RNP granule formation. Thus, one predicts that in a meg3/4 mutant, heat shock would restore P-granules. (Similar results have been seen with P-bodies in mammalian cells, where lsm4 RNAi abolishes P-bodies, but when cells are then stressed, P-bodies rapidly reassemble (Kedersha et al., 2005). Given that this is a simple experiment, and if it worked would reveal a new property of P-granules, it would be nice to try, although it is not required for publication.

We have done this experiment and found that Y51F10.2 RNA does NOT accumulate in granules in embryos depleted of MEG proteins even upon heat shock (see new panel in Figure 2—figure supplement 2D).

Similarly, ectopic localization of non-P granule RNAs to P granules upon heat shock also requires MEG proteins (Figure 2—figure supplement 2D).

We conclude from these experiments that formation of large (micron-sized) RNA granules requires MEG proteins both in wild-type and under our heat shock conditions. Our experiments, however, do not exclude the possibility that smaller RNA aggregates form in the absence of MEG proteins.

3b) A related issue is that it would improve the manuscript to clarify if separate stress granules form under heat shock, or those that form, merge with P-granules.

We have monitored the formation of stress granules using a G3BP::GFP fusion and find that G3BP puncta form in all cells under our heat shock conditions. In P blastomeres, some G3BP puncta associate with P granules. We have added these data as a new panel (Figure 2—figure supplement 2E) referred to in Materials and methods section.

4) As I read it, the analysis of mRNAs in Figure 1D that are called as "enriched" in P-granules will also be strongly affected by their abundance. As such, this analysis could miss low abundance but highly enriched mRNAs. I suggest either a) clarify how the analysis was done such that this is not an issue, or b) acknowledge the limitations in the text with how this analysis is performed and the types of RNAs it will miss.

We have added acknowledgements in Results section and Discussion section that low abundance mRNAs may have been missed.

Reviewer #2:[…] This paper presents several new and significant advances to the RNP phase-separation and germ-granule fields. Two crucial but addressable criticisms are discussed.1) As a control for the MEG-3 iCLIP experiments, the authors perform iCLIP on GFP-expressing embryos. Information for the control (JH1094?) isn't given. The optimal control would express GFP in just the germline blastomeres where MEG-3 and PGL-1 are enriched to avoid identifying the most abundant transcripts in P blastomeres (compared to the whole embryo). Genes like cbd-1, cgh-1, and puf-5 are some of most abundant in the germline. Of the 657 MEG-3-bound transcripts, over half are in the top 1000 germline expressed (11-fold more than expected by chance), and most fall within the top 3000 germline-expressed genes. The statement on line 102 that MEG-3 binds to a specific subset of mRNAs instead of abundant P-blastomere mRNAs will need more supporting data than is currently provided.Correlations provided in Figure 1C and Figure 1—figure supplement 1C may be erroneous and serve as a distraction at the beginning of an otherwise very significant study. Instead, the comparison should be with all ~ 15,000 germline-expressed transcripts. Despite a potential lack of MEG-3 specificity, results are still in line with the author's later and well-supported observations that untranslated transcripts in translationally quiescent P blastomeres have an affinity for MEG-3.

We now include the requested comparisons showing all transcripts (Figure 1—figure supplement 1C and D). These comparisons confirm that our iCLIP experiments recovered specific transcripts within a wide range of expression levels. We acknowledge, however, that low abundance transcripts may have been missed, and have added this caveat to the text in the Results section and Discussion section.

JH1094 expresses GFP in all blastomeres and therefore is not the optimal control suggested by the reviewer. Nevertheless, when we plot the abundance of all P blastomere-enriched transcripts versus their abundance in the MEG-3 IP, we obtain only a low correlation (Figure 1C), confirming that the MEG-3 IP is recovering specific transcripts.

An apparent exception to the transcript abundance correlation is that MEG-3-bound transcripts lack those that encode ribosomal proteins. While not necessary to include in this manuscript, it may be interesting to consider why abundant RPL/RPS (RPG) transcripts do not bind MEG-3.

mRNAs coding for ribosomal proteins, while abundant, are well translated, and thus were not recovered in the MEG-3 iCLIP, which generally favors mRNAs with low ribosome footprints.

2) P blastomeres are translationally quiescent, so shouldn't lower ribosomal coverage of MEG-3 attached transcripts in P blastomeres be expected when the experiment was done on whole embryos? Specificity is not addressed and observations in Figure 2C may be a correlation artifact. While this isn't likely given the later in situ hybridization experiments following translational inhibition, the caveat should be discussed.

We have modified Figure 2—figure supplement 2A to show ribosome footprints on all embryonic transcripts, transcripts enriched in P blastomeres, and P granule transcripts – this analysis confirms that P granule transcripts have on average lower ribosome coverage than even P blastomere-enriched transcripts.